# SQS: Bayesian DNN Compression through Sparse Quantized Sub-distributions

## Abstract

Compressing large-scale neural networks is essential for deploying models on resource-constrained devices. Most existing methods adopt weight pruning or low-bit quantization individually, often resulting in suboptimal compression rates to preserve acceptable performance drops. We introduce a unified framework for simultaneous pruning and low-bit quantization via Bayesian variational learning (SQS), which achieves higher compression rates than prior baselines while maintaining comparable performance. The key idea is to employ a spike-and-slab prior to inducing sparsity and model quantized weights using Gaussian Mixture Models (GMMs) to enable low-bit precision. Due to the intractability of the objective involving spike-and-slab priors with GMMs, we derive an efficient approximation that facilitates effective compression with minimal accuracy loss. In theory, we provide the consistent result of our proposed variational approach to a sparse and quantized deep neural network. Extensive experiments on compressing ResNet, BERT-base, Llama3, and Qwen2.5 models show that our method achieves higher compression rates than a line of existing methods with comparable performance drops. Code implementation of SQS and baselines are available at: `https://anonymous.4open.science/r/SQS_private-411C`.

## 1 Introduction

Deep Neural Networks (DNNs) have achieved state-of-the-art performance across a wide range of tasks but at the cost of significantly increased computational and memory requirements (Radford et al., 2018; Xu et al., 2020; Touvron et al., 2023; Kumar et al., 2025), making deployment on resource-constrained devices challenging. Model compression is thus proposed to reduce the size and computational complexity of DNNs while maintaining predictive accuracy, including pruning (LeCun et al., 1989; Han et al., 2016), weight quantization (Courbariaux et al., 2015; Rastegari et al., 2016; Frantar et al., 2023; Lin et al., 2024), knowledge distillation (Park et al., 2019; Gou et al., 2021), and neural architecture search (Liu et al., 2018; Wang et al., 2020b).

Among these, *weight pruning* and *low-bit quantization* are particularly effective and widely adopted for compressing DNNs (Buciluă et al., 2006; Choudhary et al., 2020; Liu et al., 2025a). Weight pruning eliminates redundant or unimportant weights by setting selected weights to zero, thereby reducing the number of active parameters without significantly altering the model architecture (You et al., 2019; Guo et al., 2016; Dong et al., 2017). On the other hand, quantization reduces the bit-width of numerical representations for inputs, outputs, and weights, by converting high-precision formats (e.g., `FP32`) to lower-precision alternatives, like `FP8` or `BF8`. This quantization coarsens the model representation and yields significant reductions in memory footprint and computation overhead. It enhances efficiency in both training and inference across diverse architectures, including ResNet (Banner et al., 2018), Transformers (Sun et al., 2019), Large language model (Dettmers et al., 2023; Wang et al., 2025), and vision-language models (Wortsman et al., 2023).

However, quantization and pruning inevitably introduce distributional shifts from the original DNNs, often leading to performance degradation (Dong et al., 2022). To mitigate this, existing methods adopt conservative compression rates, limiting their applicability to resource-constrained environments (Wang et al., 2020c;b;

Bai et al., 2022; Frantar & Alistarh, 2022; Bai et al., 2023). How to achieve high compression rates while maintaining acceptable performance remains an interesting question to explore.

To tackle the above problem, we introduce a unified framework: **S**parse **Q**uantized **S**ub-distribution (SQS) compression method, which unifies pruning and quantization within a single variational learning process. Instead of applying pruning and quantization separately, the key idea of SQS is joint pruning and quantization that learns a sparse, quantized sub-distribution over network weights through variational learning. To model the variational posterior, we adopt a spike-and-slab prior combined with a Gaussian Mixture Model (GMM): the spike component encourages sparsity for pruning, while the GMM component models a quantized weight distribution, effectively mitigating performance degradation. The training pipeline of SQS is in Figure 1. Theoretically, we show that under mild conditions, our SQS method finds a sparse and quantized neural network that converges to the true underlying target neural network with high probability.

In our experiments, we compare several recent state-of-the-art compression methods across a range of widely used neural networks, including ResNet, BERT-base, LLaMA3, and Qwen2.5. Our findings show that (1) Under the same bit-width setting, our SQS achieves the highest compression rate, requiring fewer parameters than baselines. (2) At the same compression rate, our SQS achieves the smallest accuracy drop or F1 score drop among all approaches, with particularly strong performance at 2-bit and 4-bit precision.

Further ablation studies highlight the contributions of individual components: (1) The spike-and-slab distribution is more effective in promoting sparsity than Gaussian alternatives. (2) Bayesian averaging during inference outperforms greedy decoding. (3) An outlier-aware window strategy better preserves informative weight outliers compared to uniform windowing, further improving performance.

## 2 Preliminaries

**Low-bit Quantization** uses discrete low-bit values to approximate full-precision floating points, primarily to reduce precision for more efficient storage and computation while preserving essential information (Gholami et al., 2022). Formally, it is defined as a mapping $Q : \theta \in \mathbb{R} \to \mathcal{Q} = \{\mu_1, \ldots \mu_K\}$, where $\theta$ is the full-precision weight and $\mathcal{Q}$ denotes the set of low-bit discrete values. Representative quantization methods include deterministic quantization (Jacob et al., 2018), stochastic quantization (Courbariaux et al., 2015), and end-to-end learnable quantization (Dong et al., 2022).

Specifically, let $\theta = (\theta_1, \ldots, \theta_T) \in \mathbb{R}^T$ represent the full-precision weights of a deep neural network, with $\theta_i$ denoting the $i$-th weight. Given a quantization set $\mathcal{Q} = \{\mu_1, \ldots, \mu_K\}$, a general stochastic quantization is a mapping $Q : \theta \to \mathcal{Q}$, which is

$$Q(\theta_i) = \mu_k, \qquad \text{with probability } p_{ki},$$

for $i = 1, \ldots, T$. Here $\mu_k$ is the learnable parameter and $p_{ki}$ is the corresponding probability for weight $\theta_i$ to be quantized to weights $\mu_k$. A key challenge is the distribution divergence between the quantized weights and the original weights, leading to significant performance degradation (Dong et al., 2022). To mitigate this, Dong et al. (2022) propose to approximate the quantized weight distribution using a Gaussian Mixture Model (GMM):

$$Q(\theta_i) \approx \sum_{k=1}^{K} \phi_k(\theta_i; \pi_k) \mathcal{N}(\mu_k, \sigma_k^2), \tag{1}$$

where $\mathcal{N}(\mu_k, \sigma_k^2)$ denotes a Gaussian distribution, and $\phi_k(\theta_i; \pi_k)$ is the mixture weight for the Gaussian components $\mathcal{N}(\mu_k, \sigma_k^2)$. To control the sharpness of this mixture, a temperature-scaled softmax is applied to obtain $\phi_k(\theta_i; \pi_k)$, that is:

$$\phi_k(\theta_i; \pi_k) = \frac{\exp\left(\varphi_k(\theta_i; \pi_k)/\tau\right)}{\sum_{j=1}^{K} \exp\left(\varphi_j(\theta_i; \pi_j)/\tau\right)}, \tag{2}$$

where the temperature parameter $\tau > 0$ controls the concentration of the distribution. As $\tau \to 0$, the GMM in Equation (1) approaches a single dominant Gaussian component. Given a prior distribution $(\pi_1, \ldots, \pi_K)$

over the quantization set $\mathcal{Q}$, the posterior component weight $\varphi_k(\theta_i; \pi_k)$ is:

$$\varphi_k(\theta_i; \pi_k) = \frac{\exp(\pi_k \mathcal{N}(\theta_i \mid \mu_k, \sigma_k^2))}{\sum_{j=1}^{K} \exp(\pi_j \mathcal{N}(\theta_i \mid \mu_j, \sigma_j^2))}.$$

Additionally, with sufficiently small $\sigma_k^2$, the GMM approximates a multinomial distribution over $\mathcal{Q}$, effectively bridging continuous and discrete quantization. For simplicity, we denote $\phi_k(\theta_i)$ as a shorthand for $\phi_k(\theta_i; \pi_k)$ throughout the remainder of this paper.

In our experiments, we find that the GMM-based compression method (Dong et al., 2022) still cannot achieve a high compression rate while maintaining low-performance drop, as it cannot efficiently encourage sparsity during training.

**Variational learning.** Given an observed dataset $D$, the goal of a Bayesian framework is to infer the true posterior distribution $\pi(\theta|D) \propto \pi(\theta)p(D; \theta)$, where $\pi(\theta)$ denotes the prior and $p(D; \theta)$ the likelihood. Since the posterior is generally intractable, variational learning (Jordan et al., 1999) is proposed to approximate it by selecting the closest distribution from a variational family $\mathcal{F}$ in terms of the Kullback–Leibler (KL) divergence (Csiszar, 1975):

$$q^*(\theta) = \underset{q(\theta) \in \mathcal{F}}{\arg\min} \, \mathrm{KL}(q(\theta) \,\|\, \pi(\theta \mid D)). \tag{3}$$

Following Blei et al. (2017), this optimization is equivalent to minimizing the negative Evidence Lower Bound (ELBO), defined as:

$$\Omega(q) - \mathbb{E}_{q(\theta)}[\log p(D; \theta)] + \mathrm{KL}(q(\theta)\|\pi(\theta)), \tag{4}$$

where the first term measures how well the variational distribution $q(\theta)$ aligns with the log-likelihood of the observed data, and the second term regularizes $q(\theta)$ to stay close to the prior $\pi(\theta)$.

Our SQS method employs a variational family based on a spike-and-GMM distribution to approximate the sparse and quantized posterior. The first term in Equation (4) allows the spike-and-GMM to learn the posterior distribution given the data. For the second term, we adopt a slack-and-slab prior $\pi(\cdot)$ distribution to promote sparsity in the network weights.

## 3 Methodology

The objective is to approximate a full-precision neural network $f(\cdot; \theta)$ with a Bayesian model $f(\cdot; \tilde{\theta})$ that is both sparse and low-precision, while minimizing performance degradation. To achieve this, we employ a spike-and-slab distribution combined with a GMM to parameterize the variational posterior.

### 3.1 SQS: Variational learning for sparse and quantized sub-distribution

The spike-and-slab prior consists of a point mass at zero (spike) and a continuous distribution (slab) (Bai et al., 2020; Ishwaran & Rao, 2005). Formally, let $\gamma = (\gamma_1, \ldots, \gamma_T)$ be a binary indicator vector, where each $\gamma_i$ determines whether the corresponding weight $\theta_i$ is preserved ($\gamma_i = 1$) or pruned ($\gamma_i = 0$). The prior for each weight $\tilde{\theta}_i$ is defined as:

$$\gamma_i \sim \mathrm{Bern}(\lambda),$$
$$\tilde{\theta}_i \mid \gamma_i \sim \gamma_i \mathcal{N}(0, \sigma_0^2) + (1 - \gamma_i)\delta_0,$$

where $\lambda$ is the prior probability of retaining a weight, and $\sigma_0^2$ is the prior variance of the Gaussian slab. Marginalizing out the binary variable $\gamma_i$, the prior distribution over $\tilde{\theta}_i$ becomes:

$$\pi(\tilde{\theta}_i) = \lambda \mathcal{N}(0, \sigma_0^2) + (1 - \lambda)\delta_0, \tag{5}$$

where $1 - \lambda$ corresponds to the prior pruning probability. For example, in a DNN with a target sparsity of 90%, setting $\lambda = 0.1$ implies that each weight has a 90% prior probability of being pruned.

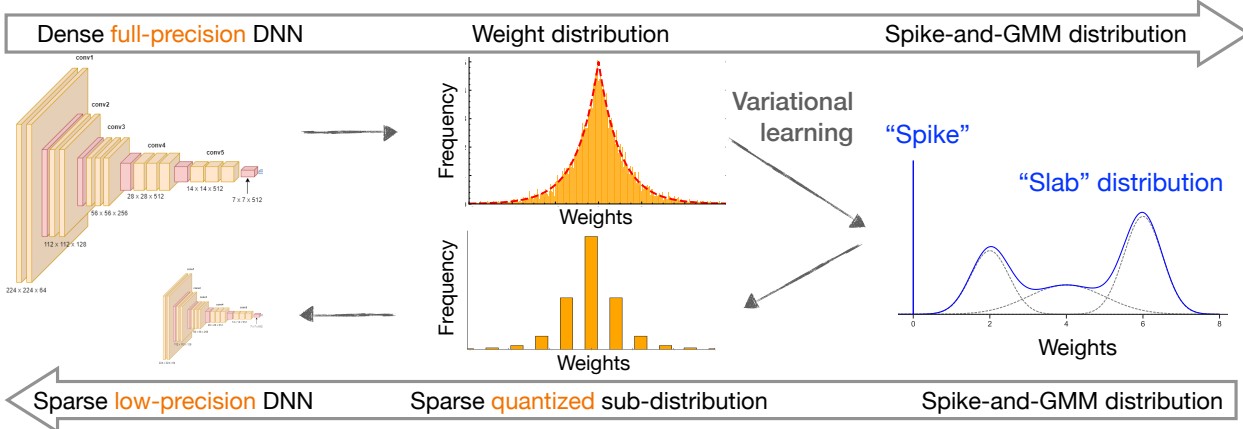

Figure 1: Our SQS method achieves high compression with minimal performance degradation by jointly pruning and quantizing model weights through variational learning. We employ a spike-and-GMM variational distribution to approximate full-precision weights: the *spike* component promotes sparsity for pruning, while the *slab* component (i.e., GMM) models a quantized weight distribution.

### 3.1.1 Training Procedure

To incorporate quantization into the variational family, we extend the spike-and-slab formulation by modeling the slab using a $K$-component GMM. Each variational distribution $q(\theta_i)$ is then defined as:

$$\gamma_i \sim \texttt{Bern}(\tilde{\lambda}_i)$$

$$\tilde{\theta}_i|\gamma_i \sim \gamma_i \sum_{k=1}^{K} \phi_k(\theta_i)\mathcal{N}(\mu_k, \sigma_k^2) + (1-\gamma_i)\delta_0$$

where $\phi_k(\theta_i)$ is the mixture weight for component $k$, and $\tilde{\lambda}_i$ is the variational probability of retaining weight $\theta_i$. The marginal variational distribution $q(\tilde{\theta}_i)$ is:

$$q(\tilde{\theta}_i) = \tilde{\lambda}_i \sum_{k=1}^{K} \phi_k(\theta_i)\mathcal{N}(\mu_k, \sigma_k^2) + (1-\tilde{\lambda}_i)\delta_0. \tag{6}$$

Given this variational family, we define the learning objective based on the ELBO:

$$\Omega(\tilde{\theta}) = -\mathbb{E}_{q(\tilde{\theta})}\big[\log p(D; \tilde{\theta})\big] + \sum_{i=1}^{T} \text{KL}\left(q(\tilde{\theta}_i)\|\pi(\tilde{\theta}_i)\right). \tag{7}$$

Yet, computing Equation (7) is intractable, as no closed-form solution exists for the KL divergence between $q(\tilde{\theta}_i)$ and the spike-and-slab prior $\pi(\tilde{\theta}_i)$. To overcome this challenge, we propose the following approximation:

$$\Omega_{\texttt{apx}}(\tilde{\theta}) = \log p\left[D; \tilde{\lambda}_i \sum_{k=1}^{K} \mu_k \phi_k(\theta_i)\right] + \sum_{i=1}^{T} \text{KL}(\texttt{Bern}(\tilde{\lambda}_i)\|\texttt{Bern}(\lambda_i)) + \sum_{i=1}^{T} \tilde{\lambda}_i \text{KL}(\mathcal{N}(\mu_{k^*}, \sigma_{k^*}^2)\|\mathcal{N}(0, \sigma_0^2)), \tag{8}$$

where $k^* = \arg\max_{1 \le k \le K} \phi_k(\theta_i)$. The first term is an approximation of the $\mathbb{E}_{q(\tilde{\theta})}[\log p(D; \tilde{\theta})]$ in Equation (7) by replacing $q(\tilde{\theta})$ with the Delta measure at the mean $\mathbb{E}[\tilde{\theta}] = \tilde{\lambda}_i \sum_{k=1}^{K} \mu_k \phi_k(\theta_i; \pi, \tau)$. The second and the third terms provide an upper bound of the term $\sum_{i=1}^{T} \text{KL}\left(q(\tilde{\theta}_i)\|\pi(\tilde{\theta}_i)\right)$ by applying Lemma 2. Please refere to Appendix A. for a detailed derivation of Equation (8).

### 3.1.2 Inference Procedure

In the inference stage, we first sample the sparse and quantized weight given the learned parameters and predict the output $\widehat{y}$ for each testing input $x$. Let $\widehat{q}(\cdot)$ denote the optimization solution of the above variational

learning, associated with the optimal parameter estimations $\{\hat{\theta}_i, \hat{\mu}_i, \hat{\sigma}_i^2, \hat{\lambda}_i\}_{i=1}^T$, and the corresponding $\widehat{\phi}_k$'s (for each $\hat{\theta}_i$) are obtained from Equation (2). Then the $i$-th quantized weight $\theta_i$ are sampled from the quantization function:

$$Q(\tilde{\theta}_i) = \hat{\mu}_k \qquad \text{with probability } \widehat{\phi}_k(\hat{\theta}_i) \tag{9}$$

Compared to sampling from $\mathcal{N}(\hat{\mu}_k, \hat{\sigma}_k^2)$, posterior sampling in Equation (9) reduces memory consumption.

To enforce sparsity, we introduce a user-specified pruning parameter, the `Non-zero` rate. Each weight $(\theta_i)$ is associated with a score $(\hat{\lambda}_i)$, which reflects the likelihood of being retained. We deterministically prune by setting the (i)-th weight to zero if $(\hat{\lambda}_i)$ is smaller than the `Non-zero`-quantile of all $(\hat{\lambda}_i)$ values; otherwise, the weight is kept unchanged. Formally,

$$\theta_i = \begin{cases} 0, & \text{if } \hat{\lambda}_i < \texttt{Non-zero} \text{ quantile of all } \hat{\lambda}_i, \\ \theta_i, & \text{otherwise.} \end{cases}$$

This deterministic rule provides exact control over the sparsity level, in contrast to stochastic pruning via posterior sampling (Bai et al., 2020; Sun et al., 2022), which does not guarantee a fixed sparsity rate and often requires an additional pruning step.

**Bayesian averaging.** Given a test input $x$, the predicted output $\hat{y}$ is computed using Bayesian averaging:

$$\hat{y} = \frac{1}{M} \sum_{m=1}^M f(x; \tilde{\theta}^m) \tag{10}$$

where $\tilde{\theta}^m$'s are $M$ many samples from the sparse quantized sub-distribution. Our ablation study (Figure 3) shows that Bayesian averaging consistently yields smaller accuracy degradation than the greedy alternative (detailed in Appendix Equation 11).

**Greedy approach** is to greedily select the most likely weight for making predictions on the test set. Specifically, for each quantized weight $\theta_i$, we choose the index $k^*$ corresponding to the highest posterior probability $\widehat{\phi}_{k^*}(\hat{\theta}_i)$. The quantized weight is then set to the mean $\mu_{k^*}$ of the selected component, and the predicted output $\hat{y}$ is computed using these selected means. Formally, this greedy inference strategy is given by:

$$\hat{y} = f(x; \hat{\mu}_{k^*}), \qquad k^* = \arg\max_k \widehat{\phi}_k(\tilde{\theta}_i), \qquad Q(\theta_i) = \mu_{k^*}. \tag{11}$$

We empirically compare the greedy inference approach with Bayesian averaging in the ablation study shown in Figure 3.

*Outlier-aware windowing.* Recent studies show that the weight distribution of large language models (LLMs) often contains significant outliers (Wei et al., 2022). To address this, we use an outlier-aware windowing strategy to enhance the performance of SQS. Specifically, the full-precision weights are partitioned into four groups using window sizes determined by a modified 1.5×Interquartile Range (IQR) rule Dekking (2005), which helps preserve large-magnitude weights during quantization. Each group is then quantized to $K$ representative values. As shown in the ablation study (Figure 2), this strategy outperforms the approach using equal-sized windows. Implementation details are provided in Appendix C, and the full procedure is summarized in Algorithm 1 .

**Remarks.** DGMS (Dong et al., 2022) adopts a mixture of Gaussians, but uses it primarily as a clustering mechanism. In contrast, our method leverages a principled Bayesian framework that supports posterior inference and enables Bayesian model averaging, enhancing robustness to quantization noise. Furthermore, by unifying pruning and quantization within a spike-and-GMM variational family, our approach creates a joint optimization space that encourages globally optimal solutions across both pruning and quantization.

### 3.1.3 Windowing strategy in quantization

We observe that weight distributions vary significantly across layers, including Gaussian and long-tailed forms. In particular, long-tailed distributions contain a small subset of weights with large magnitudes. And

---

**Algorithm 1** SQS: Variational learning and inference for sparse and quantized sub-distribution.

---

`// Variational Learning`

**Input:** Training data $D$; Full-precision weights $(\theta_1, \ldots, \theta_T)$; #components in GMM $K$; initial temperature $\tau, \tau'$; prior variance $\sigma_0^2$.

1: Initialize trainable parameters $\{(\hat{\mu}_k, \hat{\pi}_k, \hat{\sigma}_k)\}_{k=0}^K$;
2: **while** not converged **do**
3:      calculate the approximate objective $\Omega_{\texttt{apx}}$;                          ▷ in Equation (8)
4:      update learnable parameters with the stochastic gradient descent;
     **return** the sparse quantized weight sub-distribution $\widehat{q}(\tilde{\theta})$.

`// Variational Inference`

**Input:** A sparse quantized weight distribution $\widehat{q}(\tilde{\theta})$; #Bayesian average $M$; percentage of non-zero weights `Non-zero` (%).

5: **for** $n \leftarrow 1$ to $M$ **do**
6:      sample quantized weight $\tilde{\theta}^m$ from the posterior $\widehat{q}(\tilde{\theta})$;                 ▷ in Equation (9)
7:      get pruned weight $\tilde{\theta}^m$ with `Non-zero`, according to $\hat{\lambda}_i$;
8: predict output $\widehat{y}$ for each testing input $x$, using Bayesian average with $\{\tilde{\theta}^1, \ldots, \tilde{\theta}^M\}$; ▷ in Equation (10)
     **return** The set of predicted outputs.

---

the previous works (Nagel et al., 2020; Hubara et al., 2021; Frantar & Alistarh, 2022) have demonstrated that layer-wise compression methods lead to better performance.

To address performance degradation arising from such heterogeneous distributions, we extend our proposed method to support *layer-wise quantization*, where each group of weight parameters within a layer is assigned its own quantization set. This enables each layer to learn and utilize a distinct, trainable quantization set tailored to its distribution.

**Equal-size windowing.** For the equal window strategy, given a layer of weights $\theta$, we group the weights into 4 windows where each one has an equal window size $\frac{\max(\theta) - \min(\theta)}{4}$. Within each window, a $K$-components GMM is applied to approximate the weights distribution.

**Outlier-aware windowing.** For layers with long-tailed distributions, we further introduce an *outlier-aware windowing* strategy. Specifically, the weights $\theta$ in each layer are partitioned into four windows, with two dedicated to capturing the lower and upper tails of the distribution. To identify these tail regions, we apply a standard outlier detection rule based on the 5× interquartile range (IQR): let $q_1$ and $q_3$ denote the first and third quartiles of the weights, and define $\texttt{IQR} = q_3 - q_1$. The outlier-aware windows are then defined as

$$[\min(\theta), q_1 - 5 \times \texttt{IQR}], \qquad \text{and} \qquad [q_3 + 5 \times \texttt{IQR}, \max(\theta)] \tag{12}$$

Within each of the four windows in every layer, we fit a $K$-component GMM to approximate the local weight distribution.

We adopt the layer-wise quantization scheme with outlier-aware windowing in all our experiments. This approach improves the preservation of extreme values during quantization and enhances robustness across layers. An ablation study evaluating the effectiveness of outlier-aware windowing is presented in Figure 2.

### 3.2 Theoretical Justification of SQS method

For clarity, this section focuses on regression tasks with fully connected neural networks and shows that the variational posterior of sparse and quantized neural networks, i.e., the optimization of Equation (7). We show that this variational posterior converges to a true regression function under some mild conditions.

Consider a regression problem with random covariates,

$$Y_i = f_0(X_i) + \varepsilon_i, \quad \text{for } i = 1, \ldots, n, \tag{13}$$

where $f_0 : [0,1]^p \to \mathbb{R}$ is the underlying unknown true function, $X_i \sim \mathcal{U}([0,1]^p)$ sample from $p$-dimensional uniform distribution, $\epsilon_i \overset{iid}{\sim} \mathcal{N}(0, \sigma_\epsilon^2)$ is the noise term from Gaussian distribution of zero mean and variance $\sigma_\epsilon^2$. Let $P_0$ denote the true underlying probability measure of the data, and $p_0$ denote the corresponding density function. A $L$-hidden layer fully connected NNs with constant layer width $N$ and parameters $W_i \in \mathbb{R}^{N \times N}$, $b_i \in \mathbb{R}^N$ and activation function $\sigma(\cdot)$ can be defined as:

$$f_\theta(X) = W_{L+1}\sigma_{b_L}(W_L\sigma_{b_{L-1}}\ldots\sigma_{b_1}(W_1 X)) + b_{L+1}. \tag{14}$$

For simplicity, $\sigma_\epsilon$ is assumed to be known. Let $s^*$ be the "oracle" sparsity level (see Equation 18 in Appendix B for formal definition) and $H(T, s, K)$ be the set of network weight parameters such that the network has a sparsity of $s$ and shares at most $K$ distinct values. Let $P_0$ and $P_\theta$ be the true data distribution and the distribution under parameter $\theta$, respectively.

**Theorem 1.** *Let* $r_n^* = ((L+1)s^*/n)\log N + (s^*/n \log(p\sqrt{n/s^*}))$, $\varepsilon_n^* = \sqrt{r_n^*}\log^\delta(n)$ *for any* $\delta > 1$, *and* $\xi_n^* = \inf_{\theta \in H(T,s,K), \|\theta\| \le B} \|f_\theta - f_0\|_\infty^2$. *Then, under mild conditions specified in the supplementary material, with high probability:*

$$\int_{\mathbb{R}^T} d^2(P_\theta, P_0)\widehat{q}(\theta)d\theta \le C\varepsilon_n^{*2} + C'(r_n^* + \xi_n^*), \tag{15}$$

*where* $d(\cdot, \cdot)$ *denotes the Hellinger distance, and* $C$ *and* $C'$ *are some constants.*

*Sketch of Proof.* Based on prior work (Bai et al., 2020), the proof proceeds in two steps. Lemma 4 establishes a high-probability bound on the ELBO in Equation (7). Lemma 5 connects this bound to the convergence of the variational distribution toward the true full-precision posterior. Together, these results show that the variational posterior induced by our method converges to the true regression function with high probability. The full proof is in Appendix B. □

**Remark.** Similar to previous Bayesian sparse DNN results (Bai et al., 2020; Chérief-Abdellatif, 2020), the convergence rate of variational Bayes is determined by the deep neural network structure via 1) statistical estimation error $\varepsilon_n^*$, 2) variational error $r_n^*$, and 3) approximation error $\xi_n^*$. The first two positively relate to the network capacity, while the third one negatively relates to the network capacity. The estimation error $\varepsilon_n^*$ and variational error $r_n^*$ vanish as $n \to \infty$. Prior work (Beknazaryan, 2022) shows that under $B \ge 2$, $K \ge 6$, and $\beta$-Hölder smoothness of $f_0$, the approximation error $\xi_n^*$ also vanishes.

While the theoretical analysis mainly considers an $L$-hidden layer fully connected NN with constant layer width $N$, our method *SQS* is empirically validated on a variety of models such as ResNets, BERT-based models, and LLMs (refer to Section 5).

## 4 Related Works

**Weight pruning** was initially introduced by LeCun et al. (1989), with further development by Hassibi et al. (1993) through a mathematical method known as the Optimal Brain Surgeon (OBS). This approach selects weights for removal from a trained neural network using second-order information. Subsequent improvements, as indicated by studies (Dong et al., 2017; Wang et al., 2019; Singh & Alistarh, 2020), have adapted OBS for large-scale DNNs by employing numerical techniques to estimate the second-order information required by OBS. Meanwhile, Louizos et al. (2018) has introduced an $L_0$-regularized method to promote sparsity in DNNs. Frankle & Carbin (2019) established a critical insight that within a randomly initialized DNN, an optimal sub-network can be identified and extracted. Recently, Xia et al. (2024) showed that structured pruning combined with targeted retraining can significantly reduce computational costs while preserving robust performance for large language models. Concurrently, spike-and-slab distributions have been employed to promote sparsity in DNNs using Bayesian Neural Networks formulation (Deng et al., 2019; Blundell et al., 2015; Bai et al., 2020).

**Low-bit quantization.** Quantization improves DNN efficiency, particularly in resource-constrained environments (Sze et al., 2017; Frantar et al., 2023; Lin et al., 2024; 2025). Research in this field typically follows two paradigms: discontinuous-mapping and continuous-mapping quantization. Discontinuous-mapping

methods project full-precision weights onto a low-bit grid using rounding operations (Gupta et al., 2015; Hubara et al., 2018; Wu et al., 2018; Louizos et al., 2019; Courbariaux et al., 2015; De Sa et al., 2018; Marchesi et al., 1993). The non-differentiability of these mappings necessitates the use of the *straight-through estimator* (STE) for gradient approximation (Courbariaux & Bengio, 2016; Rastegari et al., 2016). However, STE-based training may introduce pseudo-gradients, leading to training instability (Yin et al., 2019). Meanwhile, many researchers propose post-training quantization methods that have limited access to the training dataset (Wang et al., 2020a; Hubara et al., 2021; Li et al., 2021; Frantar & Alistarh, 2022; Frantar et al., 2023; Lin et al., 2024).

Continuous-mapping quantization offers an alternative that avoids pseudo-gradients, leading to more stable training (Yin et al., 2019; Nielsen et al., 2025). These methods often use variational learning (Ullrich et al., 2017; Louizos et al., 2017; Shayer et al., 2018) or Markov Chain Monte Carlo techniques (Roth & Pernkopf, 2018) to approximate discrete weight distributions. However, variational methods often require manual prior specification (Ullrich et al., 2017; Louizos et al., 2017; Shayer et al., 2018), while MCMC approaches can be memory-intensive (Roth & Pernkopf, 2018). DGMS (Dong et al., 2022) addresses these limitations through automated quantization using GMMs. Our work extends DGMS by integrating pruning and quantization into a unified framework, thereby achieving higher compression rates.

**Large Language Model Compression.** Recent work on compressing large models has pursued several complementary directions. SpinQuant (Liu et al., 2025b) applies learned rotations to weights and embeddings to reduce outliers, making models easier to quantize. LeanQuant (Zhang & Shrivastava, 2025) introduces a loss-aware post-training quantization method that learns adaptive affine transformations and non-uniform quantization grids, aiming to preserve outlier-sensitive weights. EfficientXpert Zhao et al. (2025) uses LoRA-guided pruning to obtain domain-aware pruned models. Xu et al. (2025) targeted vision–language models and adaptively pruned attention heads using entropy-based effective rank and the Kolmogorov–Smirnov distance, reporting substantial FLOP reductions. In contrast, our SQS introduces a learnable codebook and a spike-and-slab posterior tailored to sparse, quantized weights in the low-precision regime.

## 5 Experiments

In this section, we show that our SQS achieves a much higher compression rate (see the second-to-last column in Tables 1-3) while incurring a comparable or smaller performance drop (see the last column in the same tables). Through ablation studies, we further validate that (1) under the same sparsity level, the spike-and-slab prior more effectively preserves model accuracy (see Table 4). (2) Under identical hyperparameter settings, we show that SQS with Bayesian averaging is better than using the greedy approach (see Figure 3).

### 5.1 Experiment settings

We evaluate all methods using two metrics: the *compression rate* and the *performance drop* (i.e., Accuracy drop or F1 score drop). The compression rate is defined as the memory footprint of the compressed over the original dense full-precision model:

$$\text{Compression rate} = \frac{32 \times \texttt{original weight count}}{\log_2 K \times \texttt{nonzero weight counts} + 32 \times K}, \tag{16}$$

where $K$ is the codebook size. `Non-zero` is the number of weights that are pruned to zero. In all our experiments, the `Non-zero` is configured as a hyperparameter to control the sparsity. In practice, for the models we study, the product "$\log_2 K \times$ nonzero weight counts" is usually much larger than $32 \times K$, because the number of nonzero weights is very large and the codebook size is small. In this regime, the memory used to store the indices dominates and the memory used by the codebook is very small.

We compare methods of different compression types (the "Compression type" column): "P+Q" denotes combined pruning and quantization, "P" denotes pruning only, and "Q" denotes quantization only.

In our experiments, we store the indices of the weights using `INT4`, and we perform computation in `FP32`. This setting is common in prior work, such as QLoRA (Dettmers et al., 2023) and learned codebook methods van den Oord et al. (2017); Dong et al. (2022).

Table 1: For compressing ResNet models, we benchmark all methods evaluated on the CIFAR-10 dataset. Using ResNet-32 and ResNet-56 models, our SQS consistently achieves higher compression rates with smaller Top-1 Accuracy drops compared to all baselines.

| | Methods | Compression type | Bits | Non-zero rate (%) | Compression rate | Top-1 Accuracy drop |
|---|---|---|---|---|---|---|
| ResNet-20 | LQNets (Zhang et al., 2018) | Q | 2 | 100% | 16× | 1.20% |
| | DGMS (Dong et al., 2022) | P+Q | 2 | 56% | 29× | **0.87**% |
| | SQS (Ours) | P+Q | 2 | **50**% | **32**× | 1.47% |

**(a)** Compressing 32Bits ResNet-20 model on CIFAR-10 dataset with Top-1 Accuracy 92.60%.

| | Methods | Compression type | Bits | Non-zero rate (%) | Compression rate | Top-1 Accuracy drop |
|---|---|---|---|---|---|---|
| ResNet-32 | TTQ (Zhu et al., 2017) | Q | 2 | 100% | 16× | 1.90% |
| | DGMS (Dong et al., 2022) | P+Q | 2 | 59% | 27× | 1.30% |
| | SQS (Ours) | P+Q | 2 | **50**% | **32**× | **1.29**% |

**(b)** Compressing 32Bits ResNet-32 model on CIFAR-10 dataset with Top-1 Accuracy 93.53%.

| | Methods | Compression type | Bits | Non-zero rate (%) | Compression rate | Top-1 Accuracy drop |
|---|---|---|---|---|---|---|
| ResNet-56 | TTQ (Zhu et al., 2017) | Q | 2 | 100% | 16× | 1.06% |
| | L1 (Li et al., 2017) | P | 32 | 10% | 10× | 1.83% |
| | DGMS (Dong et al., 2022) | P+Q | 2 | 52% | 31× | 0.89% |
| | SQS (Ours) | P+Q | 2 | **50**% | **32**× | **0.84**% |

**(c)** Compressing 32Bits ResNet-56 model on CIFAR-10 dataset with Top-1 Accuracy 94.37%.

To ensure fair comparison, each method is initialized with the same full-precision pre-trained model and is run with the same set of hyperparameters for compression. All methods are constrained to a maximum runtime of 24 hours. The resulting compressed models are then evaluated on the same test sets, and the key performance metrics are summarized in the corresponding tables. Appendix D provides detailed experimental configurations and baseline settings.

## 5.2 Experimental analysis

**Compression on ResNet models.** Table 1 summarizes the result of all methods for compressing ReNet-18, ReNet-32 and ReNet-56, evaluated on the CIFAR-10 dataset. On compressing the ResNet-18 model, our SQS attains a better compression rate than the baselines. On compressing ResNet-32 and ResNet-56 models, our SQS attains substantially higher compression rates while incurring smaller accuracy drops compared to the baselines. Optimizing pruning and quantization separately overlooks redundancies in each step; by merging them into a single optimization, we effectively eliminate these inefficiencies

**Compression on BERT-base model.** We apply our compression method to the BERT-base model (Devlin et al., 2019) and evaluate its performance on the SQuAD v1.1 dataset (Rajpurkar et al., 2016). The evaluation metrics include the F1 score drop and the compression rate. As shown in Table 2, our method achieves the lowest F1 score drop and the highest compression rate, outperforming existing methods. This demonstrates the effectiveness of our SQS method in preserving accuracy under aggressive compression.

**Compression on Llama and Qwen models.** In Table 3, we compare our method SQS with others on the SST2 task in the GLUE benchmark using Llama3.2-1B and Qwen2.5-0.5B models, as our method could preserve the weights outliers, which are crucial in maintaining the performance (Lin et al., 2024). We further observe a big performance drop of the DGMS (Dong et al., 2022) method for compressing Llama3.2-1B and Qwen2.5-0.5B, which occurs because the attention weights distribution is not Gaussian (see Figure 2(left)), and it fails to capture large magnitude weights. Furthermore, DGMS doesn't allow customization of the sparsity level, thus it presents an unreasonable performance drop.

Table 2: Compressing 32Bits Bert-base model on SQuADv1.1 dataset with F1 score 88.68%. Our SQS achieves higher compression rates with smaller F1 score drops compared to all baselines.

| | Methods | Compression type | Bits | Non-zero rate (%) | Compression rate | F1 score drop |
|---|---|---|---|---|---|---|
| BERT-base | GMP (Zhu & Gupta, 2017) | P | 32 | 50% | 2× | 22.89 |
| | L-OBS (Dong et al., 2017) | P | 32 | 50% | 2× | 10.86 |
| | ExactOBS (Frantar & Alistarh, 2022) | P | 32 | 25% | 4× | 6.43 |
| | PLATON (Zhang et al., 2022) | P | 32 | 20% | 5× | 2.20 |
| | OBQ (Frantar & Alistarh, 2022) | Q | 3 | 100% | 11× | 3.24 |
| | GPTQ (Frantar et al., 2023) | Q | 3 | 100% | 11× | 2.51 |
| | OBC (Frantar & Alistarh, 2022) | P+Q | 4 | 50% | 16× | 2.33 |
| | SQS (Ours) | P+Q | 4 | **25**% | **32×** | **1.66** |

Table 3: Compression results for LLAMA3.2 and Qwen2.5 models on the SST-2 dataset. Our SQS achieves significantly higher compression rates than AWQ while maintaining comparable ($\leq 3\%$) performance drops.

| | Methods | Compression type | Bits | Non-zero rate (%) | Compression rate | Top-1 accuracy drop |
|---|---|---|---|---|---|---|
| Llama3.2 | AWQ (Lin et al., 2024) | P | 4 | 100% | 8× | **0.46**% |
| | DGMS (Dong et al., 2022) | P+Q | 6 | 82% | 7× | 46.67% |
| | SQS (Ours) | P+Q | 6 | **25**% | **21×** | 1.48% |

**(a)** Compressing 32Bits Llama3.2-1B model on SST-2 dataset with Top-1 Accuracy 94.72%.

| | Methods | Compression type | Bits | Non-zero rate (%) | Compression rate | Top-1 accuracy |
|---|---|---|---|---|---|---|
| Qwen2.5 | AWQ (Lin et al., 2024) | P | 4 | 100% | 8× | **1.54**% |
| | DGMS (Dong et al., 2022) | P+Q | 6 | **34**% | **16×** | 50.80% |
| | SQS (Ours) | P+Q | 6 | 50% | 11× | 2.46% |

**(b)** Compressing 32Bits Qwen2.5-0.5B model on SST-2 dataset with Top-1 Accuracy 92.60%.

Table 4: Impact of the Gaussian prior and the spike-and-slab prior, for compressing a 32 bits ResNet-18 model on the CIFAR-100 dataset with Top-1 Accuracy 79.26%. The spike-and-slab prior used in our SQS consistently yields better performance than the Gaussian prior across all sparsity-level settings.

| | Bits | Non-zero (%) | Compression rate | Top-1 Accuracy drop | |
|---|---|---|---|---|---|
| | | | | Gaussian prior | Spike-and-slab prior (Ours) |
| ResNet-18 | 4 | 50% | 16× | 4.51% | **3.12**% |
| | 4 | 40% | 20× | 5.60% | **3.21**% |
| | 4 | 30% | 27× | 11.42% | **5.54**% |
| | 4 | 20% | 40× | 44.04% | **5.59**% |

## 5.3 Ablation studies

**Prior selection: Gaussian vs. spike-and-slab.** We evaluate how the choice of prior affects compression performance, comparing a Gaussian prior (Appendix D.2) to the spike-and-slab prior in Eq. (5). Specifically, we compress ResNet-18 at varying sparsity levels by representing each layer's weights with $K = 16$ components, and evaluate accuracy on CIFAR-100.

As shown in Table 4, the spike-and-slab prior consistently outperforms the Gaussian prior across all sparsity levels. The gap becomes particularly pronounced at high sparsity, where the Gaussian prior suffers substantial

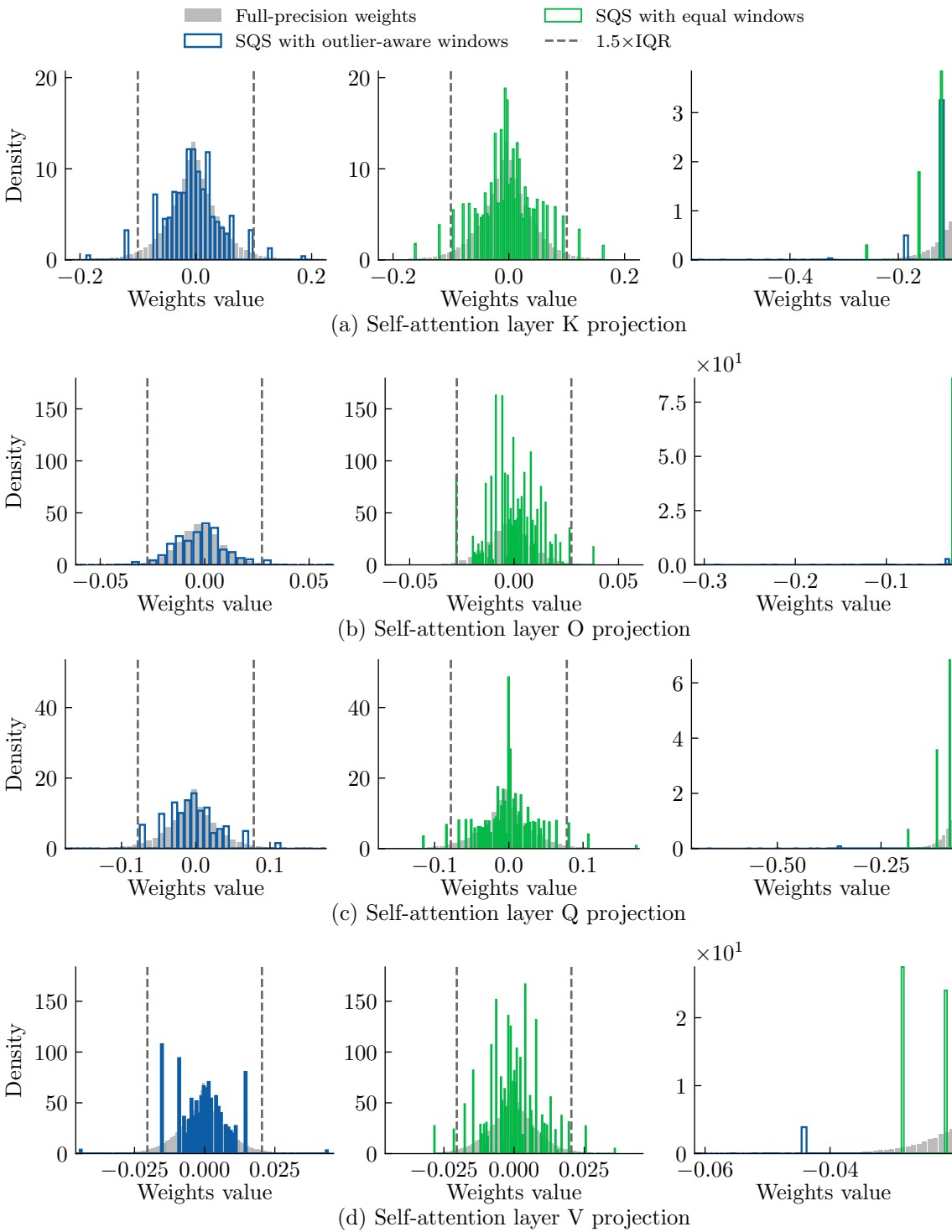

Figure 2: For the compressed weight distributions of the $K$, $O$, $Q$, and $V$ matrices in the self-attention layer of the Llama3.2-1B model, SQS using the outlier-aware window strategy (**Left**) more effectively preserves the characteristics of the full-precision weight distribution compared to the equal-sized window strategy (**Middle**). This improvement is particularly noticeable in the left tail region, as highlighted in (**Right**).

degradation, suggesting that it may be less effective at inducing posterior sparsity in DNN weights. We leave a deeper investigation of this behavior to future work.

**Windowing strategy: equal-size vs. outlier-aware window.** We use the first-layer attention weights of Llama3.2-1B as a case study. The full-precision weights exhibit a pronounced long-tail distribution, where a small fraction of entries have large magnitudes. Additional statistics on layer-wise weight distributions are reported in Appendix E. Motivated by this observation, our SQS uses an outlier-aware window strategy to better fit the full-precision weight distribution in Equation (12). Figure 2 visualizes the resulting quantized weights under the equal-window and outlier-aware window strategies.

As shown in Figure 2 (left, middle), the quantized weights produced by SQS under the outlier-aware window strategy more closely match the full-precision distribution than those obtained with an equal-window strategy. Figure 2 (right) further highlights that the outlier-aware window improves fidelity in the tails, capturing extreme-magnitude weights more accurately than equal windowing.

**Inference strategies: Bayesian averaging vs. greedy approach.** We evaluate the effectiveness of two inference strategies within our SQS framework: (1) Bayesian averaging as defined in Equation (10), (2) and the greedy approach as defined in Equation (11)) To ensure a fair comparison, we assess the performance of compressing ResNet-18 and ResNet-50 models while varying the number of Gaussian components. The sparsity level is fixed to zero (i.e., no pruning), so that all performance degradation arises purely from quantization.

As shown in Figure 3, using fewer components results in a larger accuracy drop. Under the same number of components, SQS with Bayesian averaging consistently achieves a smaller accuracy drop compared to the greedy approach.

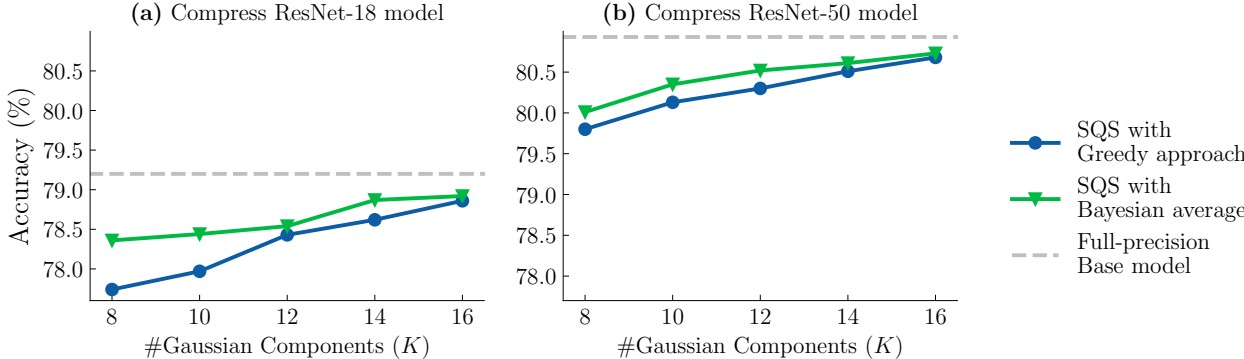

Figure 3: Comparison of inference accuracy on the CIFAR-100 dataset using ResNet-18 (left) and ResNet-50 (right). Under the same number of Gaussian components, SQS with Bayesian averaging (in Equation 10) results in a smaller accuracy drop compared to using a greedy approach (in Equation 11).

## 6 Conclusion

In this paper, we proposed a unified framework for compressing full-precision DNNs by combining pruning and quantization into one integrated optimization process through variational learning. Unlike conventional approaches that apply pruning and quantization sequentially—often resulting in suboptimal solutions—our method jointly explores a broader solution space, achieving significantly higher compression rates with comparable performance degradation. To address the intractability of the original objective, we introduce an efficient approximation that enables scalable optimization. We evaluate our method across a range of benchmarks, including ResNets, BERT-base, Llama3, and Qwen2.5. Experimental results demonstrate that our approach consistently outperforms existing baselines in compression rate while maintaining competitive accuracy, highlighting its potential for efficient deployment in resource-constrained environments.

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

## Contents

# A   Derivation of Approximate Objective

## A.1   An upper bound on the KL divergence between two mixtures

To simplify the ELBO and validate our approach, we reformulate a key lemma from previous work (Chérief-Abdellatif & Alquier, 2018, Lemma 6.1). It is a tool widely used in signal processing Hershey & Olsen (2007). We provide the proof for the sake of completeness.

**Lemma 2** (From Lemma 6.1 in Chérief-Abdellatif & Alquier (2018) ). *For any $K > 0$, the KL divergence between any two mixture densities $\sum_{k=1}^{K} w_k g_k$ and $\sum_{k=1}^{K} \tilde{w}_k \tilde{g}_k$ is upper bounded by*

$$\mathrm{KL}\left(\sum_{k=1}^{K} w_k g_k \Big\| \sum_{k=1}^{K} \tilde{w}_k \tilde{g}_k\right) \le \sum_{k=1}^{K} w_k \mathrm{KL}(g_k \| \tilde{g}_k) + \sum_{k=1}^{K} w_k \log\left(\frac{w_k}{\tilde{w}_k}\right)$$

*Proof.* We expand the KL divergence term by its definition, thus it could have the following:

$$\begin{aligned}
\mathrm{KL}\left(\sum_{k=1}^{K} w_k g_k \Big\| \sum_{k=1}^{K} \tilde{w}_k \tilde{g}_k\right) &= \int \left(\sum_{k=1}^{K} w_k g_k\right) \log\left(\frac{\sum_{k=1}^{K} w_k g_k}{\sum_{k=1}^{K} \tilde{w}_k \tilde{g}_k}\right) \\
&\le \int \sum_{k=1}^{K} w_k g_k \log\left(\frac{w_k g_k}{\tilde{w}_k \tilde{g}_k}\right) \\
&= \int \sum_{k=1}^{K} w_k g_k \log\left(\frac{g_k}{\tilde{g}_k}\right) + \int \sum_{k=1}^{K} w_k g_k \log\left(\frac{w_k}{\tilde{w}_k}\right) \\
&= \sum_{k=1}^{K} w_k \int g_k \log\left(\frac{g_k}{\tilde{g}_k}\right) + \sum_{k=1}^{K} w_k \log\left(\frac{w_k}{\tilde{w}_k}\right)\left(\int g_k\right) \\
&= \sum_{k=1}^{K} w_k \mathrm{KL}(g_k \| \tilde{g}_k) + \sum_{k=1}^{K} w_k \log\left(\frac{w_k}{\tilde{w}_k}\right),
\end{aligned}$$

where the first inequality is due to the Jensen inequality and the convexity of the function $x \log(x)$. This completes the proof. $\qquad\square$

## A.2   Derivation of Approximate Objective

We aim to approximate the ELBO objective:

$$\Omega(\tilde{\theta}) = -\underbrace{\mathbb{E}_{q(\tilde{\theta})}[\log p(D; \tilde{\theta})]}_{\text{Part 1}} + \sum_{i=1}^{T} \mathrm{KL}\left(q(\tilde{\theta}_i) \| \pi(\tilde{\theta}_i)\right). \tag{17}$$

where $\pi(\tilde{\theta}_i)$ is defined in Equation (5) and $q(\tilde{\theta}_i)$ is defined in Equation (6):

$$q(\tilde{\theta}_i) = \tilde{\lambda}_i \sum_{k=1}^{K} \phi_k(\theta_i) \mathcal{N}(\mu_k, \sigma_k^2) + (1 - \tilde{\lambda}_i)\delta_0,$$
$$\pi(\tilde{\theta}_i) = \lambda \mathcal{N}(0, \sigma_0^2) + (1 - \lambda)\delta_0.$$

It is important to note that the KL divergence between the variational distribution and the spike-and-slab prior distribution does not have a closed-form solution.

**Step 1: Approximate the expected log-likelihood.** The first term $\mathbb{E}_{q(\tilde{\theta})}[\log p(D; \tilde{\theta})]$ can be expensive to compute, due to the sampling over spike-and-slab distribution. A tractable approximation is to replace the expectation with the log-likelihood at the mean parameter: $\mathbb{E}_{q(\tilde{\theta})}[\log p(D; \tilde{\theta})] \approx \log p(D; \mathbb{E}_{q(\tilde{\theta})}[\tilde{\theta}])$. Since,

$$\mathbb{E}_{q(\tilde{\theta})}[\tilde{\theta}] = \tilde{\lambda}_i \sum_{k=1}^{K} \mu_k \phi_k(\theta_i) + (1 - \tilde{\lambda}_i) * 0 = \tilde{\lambda}_i \sum_{k=1}^{K} \mu_k \phi_k(\theta_i),$$

We then have:

$$\texttt{Part 1} \approx \log p \left( D; \tilde{\lambda}_i \sum_{k=1}^{K} \mu_k \phi_k(\theta_i) \right).$$

**Step 2: Upper Bound KL between spike-and-slab distributions.** The KL divergence between the marginal variational posterior and the prior is intractable due to the presence of both the Dirac delta and the mixture components. To upper-bound the KL divergence between them, we apply Lemma 2 by matching component structure:

$$\mathrm{KL}(q\|\pi) \leq \sum_{j=1}^{2} w_j \mathrm{KL}(g_j\|\tilde{g}_j) + \sum_{j=1}^{2} w_j \log \left( \frac{w_j}{\tilde{w}_j} \right),$$

$$\text{with} \qquad w_1 = \tilde{\lambda}_i, g_1 = \sum_{k=1}^{K} \phi_k(\theta_i) \mathcal{N}(\mu_k, \sigma_k^2), \tilde{w}_1 = \lambda, \tilde{g}_1 = \mathcal{N}(0, \sigma_0^2)$$

$$w_2 = 1 - \tilde{\lambda}_i, g_2 = \delta_0, \tilde{w}_2 = 1 - \lambda, \tilde{g}_2 = \delta_0$$

Substituting into the bound, we obtain:

$$\mathrm{KL}(q(\tilde{\theta}_i)\|\pi(\tilde{\theta}_i)) \leq \tilde{\lambda}_i \mathrm{KL} \left( \sum_{k=1}^{K} \phi_k(\theta_i) \mathcal{N}(\mu_k, \sigma_k^2) \Big\| \mathcal{N}(0, \sigma_0^2) \right) + (1 - \tilde{\lambda}_i) \underbrace{\mathrm{KL}(\delta_0\|\delta_0)}_{=0}$$

$$+ \tilde{\lambda}_i \log \frac{\tilde{\lambda}_i}{\lambda} + (1 - \tilde{\lambda}_i) \log \frac{1 - \tilde{\lambda}_i}{1 - \lambda}.$$

Combining the terms, we have:

$$\mathrm{KL}(q(\tilde{\theta}_i)\|\pi(\tilde{\theta}_i)) \leq \tilde{\lambda}_i \mathrm{KL} \left( \sum_{k=1}^{K} \phi_k(\theta_i) \mathcal{N}(\mu_k, \sigma_k^2) \Big\| \mathcal{N}(0, \sigma_0^2) \right) + \mathrm{KL}(\mathrm{Bern}(\tilde{\lambda}_i)\|\mathrm{Bern}(\lambda)),$$

Note that the first term on the right-hand side, which is the KL divergence between the GMM and the Gaussian distribution, does not have a closed form. But it can be further upper-bounded as:

$$\mathrm{KL} \left( \sum_{k=1}^{K} \phi_k(\theta_i) \mathcal{N}(\mu_k, \sigma_k^2) \Big\| \mathcal{N}(0, \sigma_0^2) \right) = \mathrm{KL} \left( \sum_{k=1}^{K} \phi_k(\theta_i) \mathcal{N}(\mu_k, \sigma_k^2) \Big\| \sum_{k=1}^{K} \phi_k(\theta_i) \mathcal{N}(0, \sigma_0^2) \right)$$

$$\leq \sum_{k=1}^{K} \phi_k(\theta_i) \mathrm{KL} \left( \mathcal{N}(\mu_k, \sigma_k^2)\|\mathcal{N}(0, \sigma_0^2) \right) + \sum_{k=1}^{K} \phi_k(\theta_i) \underbrace{\log \left( \frac{\phi_k(\theta_i)}{\phi_k(\theta_i)} \right)}_{=0}$$

$$= \underbrace{\sum_{k=1}^{K} \phi_k(\theta_i) \mathrm{KL} \left( \mathcal{N}(\mu_k, \sigma_k^2)\|\mathcal{N}(0, \sigma_0^2) \right)}_{\texttt{Part 2}},$$

where the inequality is obtained by Lemma 2. Empirically, we approximate the mixture KL by evaluating only the dominant component:

$$\texttt{Part 2} \approx \mathrm{KL}(\mathcal{N}(\mu_{k^*}, \sigma_{k^*}^2)\|\mathcal{N}(0, \sigma_0^2)), \qquad \text{where } k^* = \arg\max_k \phi_k(\theta_i)$$

where we approximate the inner sum over $k$ using the maximum-weight component, which is the $k^*$-th component.

As a small temperature is needed to avoid a flat posterior distribution, which could introduce large differences between the training phase and inference phase.

Finally, putting all approximations together, we obtain:

$$\Omega_{\mathbf{apx}}(\tilde{\theta}) = -\log p\left(D; \tilde{\lambda}_i \sum_{k=1}^{K} \mu_k \phi_k(\theta_i; \pi, \tau)\right) + \sum_{i=1}^{T} \mathrm{KL}(\mathrm{Bern}(\tilde{\lambda}_i)\|\mathrm{Bern}(\lambda)) + \sum_{i=1}^{T} \tilde{\lambda}_i \mathrm{KL}(\mathcal{N}(\mu_{k^*}, \sigma_{k^*}^2)\|\mathcal{N}(0, \sigma_0^2)),$$

where $k^* = \arg\max_{1 \leq k \leq K} \phi_k(\theta_i)$. We thus obtain the result shown in Equation (8).

# B  Proof of Theorem 1

Consider a $L$-hidden layer fully connected neural network with the ReLU activation function $\sigma_b : \mathbb{R}^d \to \mathbb{R}^d$ defined as $\sigma_b(X) = \max\{0, X - b\}$ on some dimension $d$ and parameter $b$. The number of neurons in each layer is defined as $p_i$ for $i = 1, \ldots, L$. The weights and biases are denoted as the $W_i \in \mathbb{R}^{N \times N}$ and $b_i \in \mathbb{R}^N$. Thus given the parameters $\mathbf{p} = (p_1, \cdots, p_L)$ and let $\theta$ denote the vector obtained by stacking all entries of the weight matrices $W_i$ and bias vectors $b_i$, then the fully connected network can be presented as:

$$f_\theta(X) = W_{L+1} \sigma_{b_L}(W_L \sigma_{b_{L-1}} \ldots \sigma_{b_1}(W_1 X)) + b_{L+1}.$$

The DNN $f_\theta$ also introduces a probability measure of the data, which we denote as $P_\theta$, and $p_\theta$ is the corresponding density function, $p_\theta(D)$ would be the likelihood of the data $D$.

One can define the sparse parameter space with sparsity parameter $s$ as $\{\theta \in \mathbb{R} : \|\theta\|_0 < s\}$, where $\theta$ has only $s$ many non-zero entries. Then we can further introduce the sparse and quantized weights space $H(T, s, K)$ as follows:

$$\mathcal{I}(T, s, K) = \{I = [r_1, \ldots, r_T]^\top \mid \|I\|_0 \leq s, r_i \in \{0, 1\}^K, \|r_i\|_0 \leq 1\},$$
$$H(T, s, K) = \left\{\theta \in \mathbb{R}^T \mid \theta = I \cdot E, E \in [-B, B]^K, I \in \mathcal{I}(T, s, K)\right\},$$

where $B$ is some constant satisfies $B > 2$ and $\mathcal{I}(T, s, K)$ is the indexing space, each element $I \in \mathcal{I}(T, s, K)$ consists of $s$ many $K$-dimension one-hot rows, and the rest $T - s$ rows are zero vectors indicating the corresponding weight is pruned. In such a way, any $\theta \in H(T, s, K)$ satisfies that $\|\theta\|_0 \leq s$ and $\theta$ only have $K$ many distinct entry values then the DNN $f_\theta(\cdot) = f(\cdot; \theta)$ is sparse and quantized. The following conditions are assumed, similarly to Bai et al. (2020):

**Condition B.1.** $p_i \equiv N \in \mathbb{Z}^+$ *that can depend on $n$, and* $\lim T = \infty$.

**Condition B.2.** $\sigma(x)$ *is 1-Lipschitz continuous.*

**Condition B.3.** *The hyperparameter $\sigma_0^2$ is set to be some constant, and $\lambda$ satisfies*

$$\log\left(\frac{1}{\lambda}\right) = O\left((L+1)\log N + \log(p\sqrt{n/s^*})\right)$$
$$\log\left(\frac{1}{1-\lambda}\right) = O\left(\frac{s^*}{T}\left((L+1)\log N + \log(p\sqrt{n/s^*})\right)\right)$$

**Condition B.4.** $\max\{s^* \log(p\sqrt{n/s^*}, (L+1)s^* \log N\} = o(n)$ *and* $r_n^* \asymp \xi_n^*$.

The "oracle" sparsity $s^*$ is defined in Equation (18).

**Definition 1.** *The true function $f_0$ defined in Equation 13 is $\beta$-Hölder continuous if*

$$f_0 \in \mathcal{C}_d^\beta(M) = \left\{f : [0,1]^p \to \mathbb{R} : \sum_{\alpha:|\alpha|<\beta} \|\partial^\alpha f\|_\infty + \sum_{\alpha:\alpha=\lfloor\beta\rfloor} \sup_{\substack{x,y\in[0,1]^p \\ x\neq y}} \frac{|\partial^\alpha f(x) - \partial^\alpha f(y)|}{|x-y|_\infty^{\beta-\lfloor\beta\rfloor}} \leq M\right\}.$$

*for some constant $M > 0$.*

Following previous paper (Bai et al., 2020), we define:

$$s^* = \arg\min_s \{r_n(L, p, s) + \xi_n(L, p, s)\} \tag{18}$$

where

$$r_n(L, p, s) = ((L+1)s/n)\log N + (s/n)\log(p\sqrt{n/s})$$
$$\xi_n(L, p, s) = \inf_{\theta \in H(T,s,K), \|\theta\|_\infty \leq B} \|f_\theta - f_0\|_\infty^2.$$

Correspondingly, we define $r_n^* = r_n(L, p, s^*), \xi_n^* = \xi_n(L, p, s^*)$.

In this section, we reformulate the variational distribution by introducing a latent index variable. For any $q(\tilde{\theta}) \in \mathcal{F}$, it has the following equivalent form:

$$
\begin{aligned}
\tilde{\theta}_i | z_i, \gamma_i &\sim \gamma_i \sum_{k=1}^{K} \mathbb{1}\{z_i = k\} \mathcal{N}(\mu_k, \sigma_k^2) + (1 - \gamma_i)\delta_0, \\
z_i &\sim \text{Categorical}(\phi_1(\theta_i), \phi_2(\theta_i), \ldots, \phi_K(\theta_i)), \\
\gamma_i &\sim \text{Bernoulli}(\lambda_i),
\end{aligned}
\tag{19}
$$

In addition, for theoretical convenience, we further restrict the variational family to satisfy

**Condition B.5.** $|\mu_k| \le B$ and $\sigma_k^2 \le \frac{1}{2\log(T/r_n^* \log^2(n))}$.

Note that the requirement of $|\mu_k| \le B$ is fairly reasonable, as most of the existing approximation results (Chérief-Abdellatif, 2020) only need bounded DNN weights.

We restate a formal version of our Theorem 1 as follows:

**Theorem 3.** *Under Conditions B.1-B.2 and B.4-B.5, Let $\sigma_0^2$ be a constant and $-\log \lambda = \log(T) + \delta[(L + 1)\log N + \log \sqrt{n}p]$ for any constant $\delta > 0$, Then with high probability:*

$$
\int_{\mathbb{R}^T} d^2(P_\theta, P_0)\widehat{q}(\theta)\,d\theta \le C\varepsilon_n^{*2} + C'(r_n^* + \xi_n^*),
\tag{20}
$$

*where $d(\cdot, \cdot)$ denotes the Hellinger distance, and $C$ and $C'$ are some constants.*

*Proof.* We introduce Lemma 4 and 5 to help us finish the proof.

**Lemma 4.** *Under Condition B.1-B.3, then with dominating probability,*

$$
\inf_{q(\theta)\in\mathcal{F}} \left\{ \text{KL}\left(q(\theta)\|\pi(\theta|\lambda)\right) + \int_{\mathbb{R}^T} l_n(P_0, P_\theta)q(\theta)\,d\theta \right\} \le Cn(r_n^* + \xi_n^*)
$$

*where $C$ is either some positive constant if $\lim n(r_n^* + \xi_n^*) = \infty$, or any diverging sequence if $\limsup n(r_n^* + \xi_n^*) \ne \infty$. And $l_n(P_0, P_\theta)$ is defined as:*

$$
l_n(P_0, P_\theta) = \frac{1}{2\sigma_\epsilon^2}(\|Y - f_\theta(X)\|_2^2 - \|Y - f_0(X)\|_2^2).
$$

**Lemma 5.** *Under Conditions B.1-B.4, if $\sigma_0^2$ is set to be constant and $\lambda \le T^{-1}\exp\{-Mnr_n^*/s_n\}$ for any positive diverging sequence $M \to \infty$, then with dominating probability, we have*

$$
\int_\Theta d^2(P_\theta, P_0)\widehat{q}(\theta)\,d\theta \le C\varepsilon_n^{*2} + \frac{3}{n}\inf_{q(\theta)\in\mathcal{F}}\left\{\text{KL}(q(\theta)\|\pi(\theta|\lambda)) + \int_\Theta l_n(P_0, P_\theta)q(\theta)\,d\theta\right\},
\tag{21}
$$

*where $C$ is some constant, and*

$$
\varepsilon_n^* := \varepsilon_n(L, N, s^*) = \sqrt{r_n(L, N, s^*)}\log^\delta(n), \text{ for any } \delta > 1.
$$

Then convergence in squared Hellinger distance follows directly from Lemmas 4 and 5, as the chosen value of $\lambda$ meets the necessary assumptions. $\qquad\square$

**Remark.** Compared to prior results, Lemma 4 demonstrates that a spike-and-slab prior combined with a Gaussian mixture model (GMM) with finitely many components can effectively approximate the true underlying function. In contrast, Lemma 5 establishes that the statistical estimation error of the spike-and-GMM variational distribution vanishes as the sample size $n \to \infty$.

### B.1 Proof of Lemma 4

*Proof.* Let $\theta^* = \arg\min_{\theta \in H(L,N,s,K)} \|f_\theta - f_0\|_\infty^2$. By definition, they are only $K$ many unique non-zero number in $\theta^*$, denoted as $\mu_k^* \in \mathbb{R}$, for $k = 1, \ldots, K$. In other words, for any $\theta_i^* \neq 0$, $\theta_i^*$ must choose from the quantization set $\mathcal{Q} = \{\mu_1^*, \ldots, \mu_K^*\}$, and we denote the choice index of $\theta_i^*$ from $\mathcal{Q}$ as $(i)$, (i.e. $\theta_i^* = \mu_{(i)}^*$). Now, given $\theta^*$, we construct $q^*(\tilde{\theta})$ as the following:

$$\tilde{\theta}_i | z_i^*, \gamma_i^* \sim \gamma_i^* \sum_{k=1}^K \mathbb{1}\{z_i^* = k\} \mathcal{N}(\mu_k^*, \sigma_n^2) + (1 - \gamma_i^*)\delta_0,$$

$$z_i^* \sim \text{Categorical}(\phi_1(\theta_i^*), \phi_2(\theta_i^*), \ldots, \phi_K(\theta_i^*)),$$

$$\gamma_i^* \sim \text{Bernoulli}(\psi_i^*), \qquad \psi_i^* = \mathbb{1}\{\theta_i^* \neq 0\}$$

where $\sigma_n^2 = \frac{s^*}{32n} \log(3N)^{-1}(2BN)^{-2L}\{(p + 1 + \frac{1}{BN-1}) + \frac{1}{(2BN)^2-1} + \frac{2}{(2BN-1)^2}\}^{-1}$. [1] Thus, we can have the following marginal distribution:

$$q^*(\theta) = \sum \mathbb{1}\{\gamma = \gamma^*\} \prod_{i=1}^T \gamma_i \left[\phi_{(i)}(\theta_i^*)\mathcal{N}(\mu_{(i)}^*, \sigma_n^2) + \sum_{k \neq (i)} \phi_k(\theta_i^*)\mathcal{N}(\mu_k^*, \sigma_n^2)\right] + (1 - \gamma_i^*)\delta_0$$

Next we first need to the bound $\int \|f_\theta - f_{\theta^*}\|_\infty^2 q^*(\theta)$. We can also write $\theta^* = \{W_1^*, b_1^*, \ldots, W_{L+1}^*, b_{L+1}^*\}$, then we define the following terms as:

$$\tilde{W}_l = \sup_{i,j} |W_{l,i,j} - W_{l,i,j}^*|,$$

$$\tilde{b}_l = \sup_i |b_{l,i} - b_{l,i}^*|.$$

Then, following the proof in (Chérief-Abdellatif, 2020, Proof of Theorem 7), we can have the following:

$$\int \|f_\theta - f_{\theta^*}\|_\infty^2 q^*(d\theta)$$

$$\leq 2N^{2L-2}\left(d + 1 + \frac{1}{BN-1}\right)^2 \left(\sum_{l=1}^L B^{2l-2} \prod_{v=l+1}^L \int (B + \tilde{W}_v)^2 q(d\theta) \int \tilde{W}_l^2 q_l(d\theta_l)\right.$$

$$+ 2\sum_{l=1}^L \sum_{k=1}^{l-1} B^{l-1}B^{k-1} \prod_{v=l+1}^L \int (B + \tilde{W}_v)^2 q(d\theta) \int \tilde{W}_l q_l(d\theta_l) \prod_{v=k+1}^l \int (B + \tilde{W}_v)q(d\theta) \int \tilde{W}_k q(d\theta)\right)$$

$$+ 2\left(\sum_{l=1}^L D^{2(L-l)} \prod_{v=l+1}^L \int (B + \tilde{W}_v)^2 q(d\theta) \int \tilde{b}_l^2 q(d\theta)\right.$$

$$+ 2\sum_{l=1}^L \sum_{k=1}^{l-1} D^{L-l}D^{L-k} \prod_{v=l+1}^L \int (B + \tilde{W}_v)^2 q(d\theta) \int \tilde{b}_l q(d\theta) \prod_{v=k+1}^l \int (B + \tilde{W}_v)q(d\theta) \int \tilde{b}_k q(d\theta)\right). \quad (22)$$

Then next we need to upper bound the term:

$$\int \tilde{W}_l q^*(d\theta) = \int \sup_{i,j} |W_{l,i,j} - W_{l,i,j}^*| q^*(d\theta)$$

We first bound the following, for some $t > 0$,

$$\exp\left(\mathbb{E}[t \sup_{i,j} |W_{l,i,j} - W_{l,i,j}^*|]\right) \leq \mathbb{E} \sup_{i,j} \exp\left(t|W_{l,i,j} - W_{l,i,j}^*|\right)$$

---

[1]Notice that $\sigma_n^2$ satisfies Condition B.5 as with sufficient large $n, N$ and $L$, $1/n \leq 1/\log(n)$ and $1/N^L < 1/\log(T) = 1/\log(NL)$.

$$\leq N^2 \mathbb{E}\left[\exp\left(t|W_{l,i,j} - W^*_{l,i,j}|\right)\right]$$

$$= N^2 \mathbb{E}\left[\mathbb{E}_{\mathcal{N}(\mu^*_k, \sigma^2_n)}\left[\exp(t|W_{l,i,j} - W^*_{l,i,j}|)|z_{l,i,j} = k\right]\right]$$

$$= N^2 \sum_{k=1}^{K} \phi_k(W^*_{l,i,j})\mathbb{E}_{\mathcal{N}(\mu^*_k, \sigma^2_n)}\left[\exp\left(t|W_{l,i,j} - W^*_{l,i,j}|\right)\right]$$

$$= N^2 \left(\phi_{(i)}(W^*_{l,i,j})\mathbb{E}_{\mathcal{N}(\mu^*_{(i)}, \sigma^2_n)}\left[\exp\left(t|W_{l,i,j} - W^*_{l,i,j}|\right)\right]\right.$$

$$\left. + \sum_{k\neq(i)}^{K} \phi_k(W^*_{l,i,j})\mathbb{E}\left[\exp\left(t|W_{l,i,j} - W^*_{l,i,j}|\right)\right]\right) \tag{23}$$

Notice that by definition $W^*_{l,i,j} = \mu^*_{(i)}$, thus we can bound the first term as:

$$\mathbb{E}_{\mathcal{N}(\mu^*_{(i)}, \sigma^2_n)}\left[\exp\left(t|W_{l,i,j} - W^*_{l,i,j}|\right)\right] = \mathbb{E}_{\mathcal{N}(\mu^*_{(i)}, \sigma^2_n)}\left[\exp\left(t|W_{l,i,j} - \mu^*_{(i)}|\right)\right]$$

$$= \int_0^\infty P(\exp(t|W_{l,i,j} - \mu^*_{(i)}| > x))\,dx$$

$$= \int_0^\infty P\left(|W_{l,i,j} - \mu^*_i| > \frac{\log x}{t}\right)dx$$

$$= \int_0^\infty 2P\left(W_{l,i,j} - \mu^*_{(i)} > \frac{\log x}{t}\right)dx$$

$$= \int_0^\infty 2P\left(z > \frac{\log x}{t\sigma_n}\right)dx$$

$$= 2\exp\left(\frac{t^2\sigma^2_n}{2}\right)$$

Next, we bound the second term of the Equation (23):

$$\mathbb{E}_{\mathcal{N}(\mu^*_k, \sigma^2_n)}\left[\exp(t|W_{l,i,j} - \mu^*_{(i)}|)\right] = \int_0^\infty P\left(\exp(t|W_{l,i,j} - \mu^*_{(i)}| > x)\right)dx$$

$$= \int_0^\infty 2P\left(W_{l,i,j} - \mu^*_{(i)} > \frac{\log x}{t}\right)dx$$

$$= \int_0^\infty 2P\left(W_{l,i,j} - \mu^*_k + \mu^*_k - \mu^*_{(i)} > \frac{\log x}{t}\right)dx$$

$$= \int_0^\infty 2P\left(\sigma_n z + \mu^*_k - \mu^*_{(i)} > \frac{\log x}{t}\right)dx$$

$$= 2\exp\left(t(\mu^*_k - \mu^*_{(i)}) + \frac{\sigma^2_n t^2}{2}\right)$$

$$\leq 2\exp\left(4t + \frac{\sigma^2_n t^2}{2}\right)$$

Notice that the last inequality is because of $\sup_{i,j}(\mu^*_i - \mu^*_j) \leq 4$. And by choosing $t = \sqrt{2\log(3N^2)}/\sigma_n$, the Equation (23), can be bounded by:

$$\exp\left(\mathbb{E}\left[t\sup_{i,j}|W_{l,i,j} - W^*_{l,i,j}|\right]\right) \leq 2N^2\left(\phi_{(i)}(W^*_{l,i,j})\exp(t^2\sigma^2_n/2) + \sum_{k\neq(i)}\phi_k(W^*_{l,i,j})\exp(4t + t^2\sigma^2_n/2)\right)$$

$$= 2N^2\left(\exp(t^2\sigma^2_n/2) + \sum_{k\neq(i)}\phi_k(W^*_{l,i,j})\exp(4t + t^2\sigma^2_n/2)\right)$$

$$\leq 2N^2 \Big( \exp(t^2\sigma_n^2/2) + 1/2 \exp(t^2\sigma_n^2/2) \Big)$$

$$= 3N^2 \exp\Big(t^2\sigma_n^2/2\Big)$$

The second inequality is obtained by setting $\tau$ such that $\sum_{k\neq(i)} \phi_k(W^*_{l,i,j}) \leq 1/2$. Note that given a fixed DNN structure, such a $\tau$ always exists as the $\sum_{k\neq(i)} \phi_k(W^*_{l,i,j})$ monotonically decrease to zero as $\tau$ decrease. Thus, we can have the

$$\mathbb{E}\Big[ \sup_{i,j} |W_{l,i,j} - W^*_{l,i,j}| \Big] \leq \frac{3N^2}{t} + \frac{t\sigma_n^2}{2} = \sqrt{2\sigma_n^2 \log(3N^2)} \leq \sqrt{8\sigma_n^2 \log(3N)}$$

Next we bound $\int \tilde{W}_l^2 q^*(dx)$, following similar procedure, for some $t > 0$, we can have:

$$\exp\Big( \mathbb{E}\big[ t \sup_{i,j}(W_{l,i,j} - W^*_{l,i,j})^2 \big] \Big) \leq N^2 \mathbb{E}\Big[ \exp(t(W_{l,i,j} - W^*_{i,i,j})^2) \Big]$$

$$\leq N^2 \mathbb{E}\Big[ \mathbb{E}_{\mathcal{N}(\mu_k,\sigma_n^2)}[t(W_{l,i,j} - W^*_{l,i,j})^2 | z_{l,i,j} = k] \Big]$$

$$= N^2 \Big( \phi_{(i)}(W^*_{l,i,j}) \mathbb{E}_{\mathcal{N}(\mu^*_{(i)},\sigma_n^2)}\Big[ \exp(t(W_{l,i,j} - W^*_{l,i,j})^2) \Big]$$

$$+ \sum_{k\neq(i)} \mathbb{E}_{\mathcal{N}(\mu_k,\sigma_n^2)}\Big[ \exp(t(W_{l,i,j} - W^*_{l,i,j})^2) \Big] \Big) \tag{24}$$

Note that with a slight abuse of notation, the $z_{l,i,j}$ in the above equation means the latent variable $z_i$ (introduced in (19)) corresponding to the weight $W_{l,i,j}$. The first term of Equation (24) can be bounded for $t < \frac{1}{2\sigma_n^2}$ as following:

$$\mathbb{E}_{\mathcal{N}(\mu^*_{(i)},\sigma_n^2)}\Big[ \exp\Big( t(W_{l,i,j} - W^*_{l,i,j})^2 \Big) \Big] = \mathbb{E}_{\mathcal{N}(\mu^*_{(i)},\sigma_n^2)}\Big[ \exp\Big( t(W_{l,i,j} - \mu^*_{(i)})^2 \Big) \Big] = \frac{1}{\sqrt{1 - 2t\sigma_n^2}}.$$

And the second term of Equation (24) can be bounded as:

$$\mathbb{E}_{\mathcal{N}(\mu_k,\sigma_n^2)}\Big[ \exp\Big( t(W_{l,i,j} - W^*_{l,i,j})^2 \Big) \Big] = \mathbb{E}_{\mathcal{N}(\mu_k,\sigma_n^2)}\Big[ \exp\Big( t(W_{l,i,j} - \mu^*_{(i)})^2 \Big) \Big]$$

$$= \mathbb{E}_{\mathcal{N}(\mu_k,\sigma_n^2)}\Big[ \exp\Big( t(W_{l,i,j} - \mu^*_k + \mu^*_k - \mu^*_{(i)})) \Big) \Big]$$

$$= \mathbb{E}\Big[ \exp\Big( t(\sigma_n z + \mu^*_k - \mu^*_{(i)})^2 \Big) \Big]$$

$$= \frac{1}{\sqrt{1 - 2t\sigma_n^2}} \exp\Big( \frac{(\mu^*_k - \mu^*_{(i)})^2 t}{1 - 2t\sigma_n^2} \Big)$$

$$\leq \frac{1}{\sqrt{1 - 2t\sigma_n^2}} \exp\Big( \frac{16t}{1 - 2t\sigma_n^2} \Big)$$

The last inequality is again because of the property that $\sup_{i,j} |\mu^*_i - \mu^*_j| \leq 4$. Thus Equation (24) can be bounded by:

$$\exp\Big( \mathbb{E}\big[ t \sup_{i,j}(W_{l,i,j} - W^*_{l,i,j})^2 \big] \Big) \leq N^2 \Big( \frac{1}{\sqrt{1 - 2t\sigma_n^2}} + \sum_{k\neq(i)} \phi_k(W^*_{l,i,j}) \frac{1}{\sqrt{1 - 2t\sigma_n^2}} \exp\Big( \frac{16t}{1 - 2t\sigma_n^2} \Big) \Big)$$

$$\leq 2N^2 \frac{1}{\sqrt{1 - 2t\sigma_n^2}},$$

where the last inequality is because of choosing a small $\tau$ such $\sum_{k\neq(i)} \phi_k(W^*_{l,i,j}) \leq \exp(\frac{16t}{1-2t\sigma_n^2})$, such $\tau$ always exist since $\sum_{k\neq(i)} \phi_k(W^*_{l,i,j})$ monotonically decrease to 0 as $\tau \to 0$.

Thus we can bound $\int \tilde{W}_l^2 q^*(dx)$ by the following:

$$\int \tilde{W}_l^2 q^*(dx) = \int \sup_{i,j}(W_{l,i,j} - W_{l,i,j}^*)^2\, dx \leq \log(N^2)/t + \log\left(\frac{2}{\sqrt{1-2t\sigma_n^2}}\right)/t$$

$$= 4\log(N^2)\sigma_n^2 + 4\log\left(\frac{2}{\sqrt{\frac{1}{2}}}\right)\sigma_n^2$$

$$\leq 8\sigma_n^2\log(3N)$$

By choosing $\sqrt{8\sigma_n^2\log(3N)} \leq B$, we can have:

$$\int \left(B + \tilde{W}_l\right)q^*(d\theta) \leq 2B,$$

$$\int \left(B + \tilde{W}_l\right)^2 q^*(d\theta) \leq B^2 + 2B\sqrt{8\sigma_n^2\log(3N)} + 8\sigma_n^2\log(3N) \leq 4B^2.$$

Similarly, we can have the following:

$$\int \tilde{b}_l q^*(d\theta) \leq \sqrt{8\sigma_n^2\log(3N)},$$

$$\int \tilde{b}_l^2 q^*(d\theta) \leq 8\sigma_n^2\log(3N).$$

Combined with Equation (22), we can have:

$$\int \|f_\theta - f_{\theta^*}\|_\infty^* q^*(d\theta) \leq 2N^{2L-2}\left(p + 1 + \frac{1}{BN-1}\right)^2\left(\sum_{\ell=1}^{L} B^{2\ell-2}(4B^2)^{L-\ell}8\sigma_n^2\log(3N)\right.$$

$$+ 2\sum_{\ell=1}^{L}\sum_{k=1}^{\ell-1} B^{\ell-1}B^{k-1}(4B^2)^{L-\ell}\sqrt{8\sigma_n^2\log(3N)}(2B)^{\ell-k}\sqrt{8\sigma_n^2\log(3N)}\right)$$

$$+ 2\left(\sum_{\ell=1}^{L} N^{2(L-\ell)}(4B^2)^{L-\ell}8\sigma_n^2\log(3N)\right.$$

$$+ 2\sum_{\ell=1}^{L}\sum_{k=1}^{\ell-1} N^{L-\ell}N^{L-k}(4B^2)^{L-\ell}\sqrt{8\sigma_n^2\log(3N)}(2B)^{\ell-k}\sqrt{8\sigma_n^2\log(3N)}\right).$$

With some algebra, we can have:

$$\int \|f_\theta - f_{\theta^*}\|_2^2 q_n^*(d\theta)$$

$$\leq 2N^{2L-2}\left(d + 1 + \frac{1}{BN-1}\right)^2\left(B^{2L-2}8\sigma_n^2\log(3N)\sum_{\ell=0}^{L-1}4^\ell + 2B^{2L-2}8\sigma_n^2\log(3D)\sum_{\ell=1}^{L}\sum_{k=1}^{\ell-1}2^{L-\ell}2^{L-k}\right)$$

$$+ 2\left(8\sigma_n^2\log(3N)\sum_{\ell=1}^{L}(2BN)^{2L-2\ell} + 16\sigma_n^2\log(3N)\sum_{\ell=1}^{L}\sum_{k=1}^{\ell-1}(2BN)^{L-\ell}(2BN)^{L-k}\right)$$

$$\leq 2N^{2L-2}\left(p + 1 + \frac{1}{BN-1}\right)^2\left(B^{2L-2}8\sigma_n^2\log(3N)\frac{4^L-1}{4-1} + 2B^{2L-2}8\sigma_n^2\log(3N)\sum_{\ell=1}^{L}2^{L-\ell}2^{L-\ell+1}\sum_{k=0}^{\ell-2}2^k\right)$$

$$+ 2\left(8\sigma_n^2\log(3N)\sum_{\ell=0}^{L-1}(2BN)^{2\ell} + 16\sigma_n^2\log(3N)\sum_{\ell=1}^{L}(2BD)^{L-\ell}(2BN)^{L-\ell+1}\sum_{k=0}^{\ell-2}(2BN)^k\right)$$

$$\leq 2N^{2L-2}\left(d + 1 + \frac{1}{BN-1}\right)^2\left(B^{2L-2}8\sigma_n^2\log(3N)\frac{4^L}{3} + 2B^{2L-2}8\sigma_n^2\log(3N)\sum_{\ell=1}^{L}2^{L-\ell}2^{L-\ell+1}2^{\ell-1}\right)$$

$$+ 2\left(8\sigma_n^2 \log(3N)\frac{(2BN)^{2L}}{(2BN)^2 - 1} + 16\sigma_n^2 \log(3N)\sum_{\ell=1}^{L}(2BN)^{L-\ell}(2BN)^{L-\ell+1}\frac{(2BN)^{\ell-1}}{2BN - 1}\right)$$

$$\leq 2N^{2L-2}\left(d + 1 + \frac{1}{BN - 1}\right)^2\left(B^{2L-2}8\sigma_n^2\log(3N)\frac{4^L}{3} + 2B^{2L-2}8\sigma_n^2\log(3N)2^L\sum_{\ell=0}^{L-1}2^\ell\right)$$

$$+ 2\left(8\sigma_n^2\log(3N)\frac{(2BN)^{2L}}{(2BN)^2 - 1} + 16\sigma_n^2\log(3N)\sum_{\ell=0}^{L-1}(2BN)^\ell\frac{(2BN)^L}{2BN - 1}\right)$$

$$\leq 2N^{2L-2}\left(p + 1 + \frac{1}{BN - 1}\right)^2\left(B^{2L-2}8\sigma_n^2\log(3N)\frac{4^L}{3} + 2B^{2L-2}8\sigma_n^2\log(3N)2^{2L}\right)$$

$$+ 2\left(8\sigma_n^2\log(3N)\frac{(2BN)^{2L}}{(2BN)^2 - 1} + 16\sigma_n^2\log(3N)\frac{(2BN)^{2L}}{(2BN - 1)^2}\right)$$

$$= 2N^{2L-2}\left(p + 1 + \frac{1}{BN - 1}\right)^2 8\sigma_n^2\log(3N)\left(B^{2L-2}\frac{4^L}{3} + 2B^{2L-2}2^{2L}\right)$$

$$+ 2\left(\frac{(2BN)^{2L}}{(2BN)^2 - 1} + 2\frac{(2BN)^{2L}}{(2BN - 1)^2}\right)8\sigma_n^2\log(3N),$$

as $BN > 2$,

$$\int \|f_\theta - f_{\theta^*}\|_\infty^2 q^*(d\theta)$$

$$\leq 16\sigma_n^2\log(3N)\left\{N^{2L-2}\left(p + 1 + \frac{1}{BN - 1}\right)^2\frac{7}{3}B^{2L-2}2^{2L} + (2BN)^{2L}\left(\frac{1}{(2BN)^2 - 1} + \frac{2}{(2BN - 1)^2}\right)\right\}$$

$$= 16\sigma_n^2\log(3N)\left\{(2BN)^{2L}\frac{1}{(BN)^2}\left(p + 1 + \frac{1}{BN - 1}\right)^2\frac{7}{3} + (2BN)^{2L}\left(\frac{1}{(2BN)^2 - 1} + \frac{2}{(2BN - 1)^2}\right)\right\}$$

$$\leq 16\sigma_n^2\log(3N)(2BN)^{2L}\left\{\left(p + 1 + \frac{1}{BN - 1}\right)^2 + \frac{1}{(2BN)^2 - 1} + \frac{2}{(2BN - 1)^2}\right\}$$

$$= \frac{s^*}{2n} \leq r_n^*$$

The last equality is due to the definition of $\sigma_n$, and the last inequality is due to the definition of $r_n^*$.

In the next step, we aim to bound the integral $\int_{\mathbb{R}^T} l_n(P_0, P_\theta)d\theta$. Note that by definition:

$$l_n(P_0, P_\theta) = \frac{1}{2\sigma_\epsilon^2}(\|Y - f_\theta(X)\|_2^2 - \|Y - f_0(X)\|_2^2)$$

$$= \frac{1}{2\sigma_\epsilon^2}(\|Y - f_0(X) + f_0(X) - f_\theta(X)\|_2^2 - \|Y - f_0(X)\|_2^2)$$

$$= \frac{1}{2\sigma_\epsilon^2}(\|f_\theta(X) - f_0(X)\|_2^2 + 2\langle Y - f_0(X), f_0(X) - f_\theta(X)\rangle),$$

We can define the following:

$$\mathcal{R}_1 = \int_{\mathbb{R}^T}\|f_\theta(X) - f_0(X)\|_2^2 q^*(\theta)(d\theta),$$

$$\mathcal{R}_2 = \int_{\mathbb{R}^T}\langle Y - f_0(X), f_0(X) - f_\theta(X)\rangle q^*(\theta)(d\theta).$$

Since $\|f_\theta(X) - f_0(X)\|_2^2 \leq n\|f_\theta - f_0\|_\infty^2 \leq n(r_n^* + \|f_{\theta^*} - f_0\|_\infty^2)$, it follows that

$$\mathcal{R}_1 \leq nr_n^* + n\|f_{\theta^*} - f_0\|_\infty^2.$$

Given $Y - f_0(X) = \epsilon \sim \mathcal{N}(0, \sigma_\epsilon^2 I)$, we have

$$\mathcal{R}_2 = \epsilon^T\int_\Theta(f_0(X) - f_\theta(X))q^*(\theta)(d\theta) \sim \mathcal{N}(0, c_f\sigma_\epsilon^2),$$

whereby the Cauchy-Schwarz inequality,

$$c_f = \| \int_\Theta (f_0(X) - f_\theta(X))q^*(\theta)(d\theta)\|_2^2 \le \mathcal{R}_1$$

Thus, $\mathcal{R}_2 = O_p(\sqrt{\mathcal{R}_1})$, and with high probability, $\mathcal{R}_2 \le C_0'\mathcal{R}_1$ for some positive constant $C_0'$ if $\lim n(r_n^*+\xi_n^*) = \infty$, or for any diverging sequence $C_0'$ if $\limsup n(r_n^* + \xi_n^*) \ne \infty$. Therefore,

$$\int_{\mathbb{R}^T} l_n(P_0, P_\theta)q^*(\theta)(d\theta) \le C_1'(nr_n^* + \|f_{\theta^*} - f_0\|_\infty^2) \quad \text{w.h.p.} \tag{25}$$

In the next step, we try to bound the KL divergence between $q^*$ and $\pi(\theta|\lambda)$,

$$\mathrm{KL}\Big(q^*(\theta)\|\pi(\theta|\lambda)\Big)$$

$$\le \log\left(\frac{1}{\pi(\gamma^*)}\right) + \sum_{i=1}^T \mathrm{KL}\left[\gamma_i^*\left[\sum_k \phi_k(\theta_i^*)\mathcal{N}(\mu_k^*, \sigma_n^2)\right] + (1-\gamma_i^*)\delta_0 \Big\| \gamma_i^*\mathcal{N}(0,\sigma_0^2) + (1-\gamma_i^*)\delta_0\right] \tag{26}$$

$$= \log\frac{1}{\lambda^{s^*}(1-\lambda)^{T-s^*}} + \sum_{i=1}^T \gamma_i^*\mathrm{KL}\left[\sum_k \phi_k(\theta_i^*)\mathcal{N}(\mu_k^*,\sigma_n^2)\|\mathcal{N}(0,\sigma_n^2)\right]$$

$$\le s^*\log\left(\frac{1}{\lambda}\right) + (T-s^*)\log\left(\frac{1}{1-\lambda}\right) + \sum_{i=1}^T \gamma_i^*\sum_{k=1}^K \phi_k(\theta_i^*)\mathrm{KL}(\mathcal{N}(\mu_k^*,\sigma_n^2)\|\mathcal{N}(0,\sigma_0^2)) \tag{27}$$

$$\le s^*\log\left(\frac{1}{\lambda}\right) + (T-s^*)\log\left(\frac{1}{1-\lambda}\right) + \sum_{i=1}^T \gamma_i^*\sum_{k=1}^K \phi_k(\theta_i^*)\left[\frac{1}{2}\log\frac{\sigma_0^2}{\sigma_n^2} + \frac{\sigma_n^2 + (\mu_k^*)^2}{2\sigma_0^2} - \frac{1}{2}\right]$$

$$\le s^*\log\left(\frac{1}{\lambda}\right) + (T-s^*)\log\left(\frac{1}{1-\lambda}\right) + \sum_{i=1}^T \gamma_i^*\left[\frac{1}{2}\log\frac{\sigma_0^2}{\sigma_n^2} + \frac{\sigma_n^2 + 4}{2\sigma_0^2} - \frac{1}{2}\right]$$

$$\le C_0 nr_n^* + \frac{s^*}{2}\sigma_n^2 + \frac{s^*}{2\sigma_0^2}(B^2-1) + \frac{s^*}{2}\log\left(\frac{\sigma_0^2}{\sigma_n^2}\right) \tag{28}$$

$$\le (C_0+1)nr_n^* + \frac{s^*}{2\sigma_0^2}B^2 + \frac{s^*}{2}\log\Big(\frac{16n}{s^*}\log(3pN)(2BN)^{2L+2}\Big\{(p+1+\frac{1}{BN-1})^2 + \frac{1}{(2BN)^2-1} + \frac{2}{(2BN-1)^2}\Big\}\Big)$$

$$\le (C_0+2)nr_n^* + \frac{s^*}{2\sigma_0^2}B^2 + (L+1)s^*\log(2BN) + \frac{s^*}{2}\log\log(3BN) + \frac{s^*}{2}\log\Big(\frac{n}{s^*}p^2\Big)$$

$$\le (C_0+3)nr_n^* + (L+1)s^*\log N + s^*\log\Big(p\sqrt{\frac{n}{s^*}}\Big)$$

$$\le C_1 nr_n^*, \text{ for sufficiently large } n.$$

where the inequality (26) and (27) are due the Lemma 2 and the inequality (28) is because of the fact that $B > 2$ and $\sum_{i=1}^T \gamma_i^* = s^*$, thus combined with the result in equation (25), we finish the proof. $\qquad\square$

### B.2 Proof of Lemma 5

*Proof.* Following previous work (Bai et al., 2020, proof of Lemma 4.2), we first define the space

$$
\begin{aligned}
H_n(\theta) &= \{\theta \in \mathbb{R}^T : \|\theta\|_0 \leq s_n, \|\theta\|_\infty \leq B+1\}, \\
H'_n(\theta) &= \{\theta \in \mathbb{R}^T : \|\theta\|_0 > s_n, \|\theta\|_\infty \leq B+1\}, \\
H''_n(\theta) &= \{\theta \in \mathbb{R}^T : \|\theta\|_\infty > B+1\}.
\end{aligned}
$$

By the above definitions, we now have:

$$
\int_{\mathbb{R}^T} d^2(P_\theta, P_0)\widehat{q}(d\theta) = \int_{H_n(\theta)} d^2(P_\theta, P_0)\widehat{q}(d\theta) + \int_{H'_n(\theta)} d^2(P_\theta, P_0)\widehat{q}(d\theta) + \int_{H''_n(\theta)} d^2(P_\theta, P_0)\widehat{q}(d\theta). \tag{29}
$$

Lemma 6 presents a variational characterization of the KL divergence, originally due to Donsker and Varadhan. The proof is available in Boucheron et al. (2013).

**Lemma 6.** *Let $\mu$ be any probability measure and $h$ a measurable function with $e^h \in L_1(\mu)$, then*

$$
\log \int e^{h(\eta)}\mu(d\eta) = \sup_\rho \left[ \int h(\eta)\rho(d\eta) - \mathrm{KL}(\rho\|\mu) \right].
$$

We can define the truncation of distribution $\widehat{q}(\cdot)$ on the set $H_n(\theta)$ denoted as $\breve{q}(\cdot)$, (i.e. $\breve{q}(\theta) = \widehat{q}(\theta)\mathbb{1}\{\theta \in H_n(\theta)\}/\widehat{q}(H_n(\theta))$ ), similarly we can also define the $\tilde{\pi}(\theta) = \pi(\theta)\mathbb{1}\{\theta \in H_n(\theta)\}/\widehat{q}(H_n(\theta))$. By adopting the arguments from Bai et al. (2020), and following steps analogous to those leading to Equation (17) therein, we obtain:

$$
\int_{H_n(\theta)} \eta(P_\theta, P_0)\,\widetilde{\pi}(\theta)\,d\theta \leq \exp\left(C_0 n\varepsilon_n^{*2}\right), \quad \text{w.h.p.} \tag{30}
$$

for some constant $C_0 > 0$, where $\log \eta(P_\theta, P_0) = l_n(P_\theta, P_0) + \frac{n}{3}d^2(P_\theta, P_0)$.

Then, given the Lemma 4 and equation (30), we can show that the first term can be bounded w.h.p. as:

$$
\begin{aligned}
\frac{n}{3\widehat{q}(H_n(\theta))} &\int_{H_n(\theta)} d^2(P_\theta, P_0)\widehat{q}(\theta)\,d\theta \\
&\leq Cn\varepsilon_n^{*2} + \mathrm{KL}(\breve{q}(\theta)\|\widetilde{\pi}(\theta)) - \int_{H_n(\theta)} l_n(P_\theta, P_0)\breve{q}(\theta)\,d\theta \\
&= Cn\varepsilon_n^{*2} + \frac{1}{\widehat{q}(H_n(\theta))} \left( \mathrm{KL}(\widehat{q}(\theta)\|\pi(\theta)) - \int_\Theta l_n(P_\theta, P_0)\widehat{q}(\theta)\,d\theta \right) \\
&\quad - \frac{1}{\widehat{q}(H_n(\theta))} \left( \int_{H_n(\theta)^c} \log \frac{\widehat{q}(\theta)}{\pi(\theta)}\widehat{q}(\theta)\,d\theta - \int_{H_n(\theta)^c} l_n(P_\theta, P_0)\widehat{q}(\theta)\,d\theta \right) + \log \frac{\pi(H_n(\theta))}{\widehat{q}(H_n(\theta))}.
\end{aligned} \tag{31}
$$

Additionally, notice that:

$$
\int_{\theta \in H'_n(\theta)} d^2(P_\theta, P_0)\widehat{q}(\theta)\,d\theta \leq \int_{\theta \in H'_n(\theta)} \widehat{q}(\theta)\,d\theta = \widehat{q}(H'_n(\theta)),
$$

combined with the fact that $d^2(P_\theta, P_0) \leq 1$, the second term of the Equation (29) can be bounded with high probability as:

$$
\begin{aligned}
\int d^2(P_\theta, P_0)\widehat{q}(\theta)\,d\theta \leq\ & 3\widehat{q}(\Theta_n)C\varepsilon_n^{*2} + \frac{3}{n} \left( \mathrm{KL}(\widehat{q}(\theta)\|\pi(\theta)) - \int_{H_n(\theta)} l_n(P_\theta, P_0)\widehat{q}(\theta)\,d\theta \right) \\
& + \frac{3}{n} \int_{H'_n(\theta)^c} l_n(P_\theta, P_0)\widehat{q}(\theta)\,d\theta + \frac{3}{n} \int_{H'_n(\theta)^c} \log \frac{\pi(\theta)}{\widehat{q}(\theta)}\widehat{q}(\theta)\,d\theta \\
& + \frac{3\widehat{q}(H_n(\theta))}{n} \log \frac{\pi(H_n(\theta))}{\widehat{q}(H_n(\theta))} + \widehat{q}(H'_n(\theta)) + \widehat{q}(H''_n(\theta)).
\end{aligned} \tag{32}
$$

Following the procedure in (Bai et al., 2020, Lemma 4.2, equation 20), we can show with high probability that:

$$\int d^2(P_\theta, P_0)\widehat{q}(\theta)\,d\theta \le 3\widehat{q}(H_n)C\varepsilon_n^{*2} + \frac{3}{n}\left(\mathrm{KL}(\widehat{q}(\theta)\|\pi(\theta)) - \int l_n(P_\theta, P_0)\widehat{q}(\theta)\,d\theta\right) + \frac{3}{n}\int_{H_n^c} l_n(P_\theta, P_0)\widehat{q}(\theta)\,d\theta$$

$$+ \frac{3}{n}\int_{H_n^c}\log\frac{\pi(\theta)}{\widehat{q}(\theta)}\widehat{q}(\theta)\,d\theta + \frac{3\widehat{q}(H_n)}{n}\log\frac{\pi(H_n)}{\widehat{q}(H_n)} + \widehat{q}(H_n'(\theta)) + \widehat{q}(H_n''(\theta))$$

$$= 3\widehat{q}(H_n)C\varepsilon_n^{*2} + \frac{3}{n}\left(\mathrm{KL}(\widehat{q}(\theta)\|\pi(\theta)) - \int_{\mathbb{R}^T} l_n(P_\theta, P_0)\widehat{q}(\theta)\,d\theta\right) + \frac{3}{n}\int_{H_n^c} l_n(P_\theta d, P_0)\widehat{q}(\theta)\,d\theta$$

$$+ \frac{3}{n}\int_{H_n^c}\log\frac{\pi(\theta)}{\widehat{q}(\theta)}\widehat{q}(\theta)\,d\theta + \frac{3\widehat{q}(H_n)}{n}\log\frac{\pi(H_n)}{\widehat{q}(H_n)} + \widehat{q}(H_n') + \widehat{q}(H_n'')$$

$$\le C\varepsilon_n^{*2} + \frac{3}{n}\left(\mathrm{KL}(\widehat{q}(\theta)\|\pi(\theta)) - \int_{\mathbb{R}^T} l_n(P_\theta, P_0)\widehat{q}(\theta)\,d\theta\right) + O(1/n) + \widehat{q}(H_n''),$$

where $C$ is some constant. Next, we show that $\widehat{q}(H_n''(\theta)) = O(\varepsilon_n^{*2})$.

**Lemma 7.** *Given the Condition B.5, $\widehat{q}(H_n''(\theta)) = O(\varepsilon_n^{*2})$ holds.*

*Proof.* Let $\widehat{\sigma}_m = \max\{\widehat{\sigma}_1, \ldots, \widehat{\sigma}_K\}$, then by definition of the variational distribution $q(\theta)$, we can know that:

$$\widehat{q}(H_n''(\theta)) \le \sum_{i=1}^T \widehat{q}(|\theta_i| > (B+1)) = \sum_{i=1}^T \widehat{q}(\theta_i > (B+1)) + \widehat{q}(\theta_i < -(B+1)).$$

By the definition of variational distribution $\widehat{q}(\cdot)$, we know:

$$\widehat{q}(\theta_i > B+1) \le \int_{B+1}^\infty \frac{1}{\sqrt{2\pi\sigma_m^2}}\exp\left(\frac{-(t-B)^2}{2\sigma_m^2}\right)dt,$$

$$\widehat{q}(\theta_i < -(B+1)) \le \int_{-\infty}^{-(B+1)} \frac{1}{\sqrt{2\pi\sigma_m^2}}\exp\left(\frac{-(t-B)^2}{2\sigma_m^2}\right)dt.$$

And by Chernoff bound and the fact that $\sigma_m^2 \le \frac{1}{2\log(T/\varepsilon_n^{*2})}$, we can have:

$$\int_{B+1}^\infty \frac{1}{\sqrt{2\pi\sigma_m^2}}\exp\left(\frac{-(t-B)^2}{2\sigma_m^2}\right)dt \le \frac{1}{2}\exp\left(-\frac{1}{2\sigma_m^2}\right) \le \frac{\varepsilon_n^{*2}}{2T}.$$

Thus we show that $\widehat{q}(H_n'') \le \varepsilon_n^{*2}$. $\qquad\qquad\square$

Then by Lemma 7, we complete the proof. $\qquad\qquad\square$

## C   Implementation of SQS

**Pretrained model setting.** Taking the compression of the Llama3.2-1B model as an example, we first download the pre-trained model from Hugging Face[2] using the Python package "`transformers`". We then fine-tune the model on the considered SST-2 task before applying our SQS for compression. We find that omitting the fine-tuning step significantly degrades the performance of SQS.

For ResNet models, we use publicly available pre-trained models obtained by the Python package "`timm`" on the CIFAR-10 and CIFAR-100 datasets and directly apply our SQS for compression.

For the Bert-base model, we download the pre-trained model from Hugging Face[3] using the Python package "`transformers`".

**Initialization.** To initialize the learnable parameters of our SQS method, denoted as $\{\mu_k, \sigma_k, \pi_k\}_{k=1}^K$, we employ the K-means algorithm. Specifically, the DNN weights of a given layer are first clustered into $K$ groups. For each group $G_k$, the mean $\mu_k$ and standard deviation $\sigma_k$ are computed as the empirical statistics of the weights in that group, while the mixture coefficient $\pi_k$ is set to the proportion of weights in group $k$ relative to the total number of weights in the layer.

We assume that K-means yields $K$ disjoint groups of weights, denoted as $\{G_1, \ldots, G_K\}$, such that $\bigcup_{k=1}^K G_k$ covers all weights in the selected layer. The initial parameters are then defined as:

$$\mu_k = \frac{1}{|G_k|} \sum_{\theta_i \in G_k} \theta_i, \qquad \sigma_k = \sqrt{\frac{1}{|G_k| - 1} \sum_{\theta_i \in G_k} (\theta_i - \mu_k)^2}, \qquad \pi_k = \frac{|G_k|}{\sum_{j=1}^K |G_j|}.$$

**Implementing marginal** $q(\theta_i)$**.** To make the distribution differentiable, we reparameterize $\tilde{\lambda}_i$ (as defined in Equation 6) using the following equation:

$$\tilde{\lambda}_i = \frac{\exp(\tilde{s}_i/\tau')}{1 + \exp(\tilde{s}_i/\tau')}, \tag{33}$$

where the temperature $\tau'$ is set to a fixed constant to stabilize the training process. To better exploit the learned pruning parameters in later stages of training, we halve $\tau'$ after completing half of the total training steps to stabilize the training and exploit more around the optimal.

**Hyperparameter Configuration.** The number of Gaussian components is set to $K = 16$, which balances compression rate and performance degradation. This setting is further analyzed in the case study shown in Figure 3. All models are trained on an NVIDIA H100 GPU with 80GB of memory.

During training and testing, for ResNet-18, ResNet-20, ResNet-32, ResNet-50, ResNet-56, BERT-base, LLaMA3.2-1B, and Qwen2.5-0.5B models, the settings are:

- **Training Time:** Approximately 30 minutes for ResNet models (ResNet-18 through ResNet-56); 4 hours for BERT-base; 24 hours for both LLaMA3.2-1B and Qwen2.5-0.5B.

- **Optimizer:** AdamW is used consistently across all models.

- **Quantization Learning Rate:** $5 \times 10^{-4}$ for ResNet-18; $5 \times 10^{-5}$ for all other models.

- **Pruning Learning Rate:** Fixed at 0.012 for all models.

- **Temperature Hyperparameters:** $\tau$ in Eq. 6 is set to $5 \times 10^{-4}$ and $\tau'$ in Eq. 33 is set to 0.0125.

- **Pruning Schedule:** A polynomial schedule is used for all models.

---

[2]`https://huggingface.co/meta-llama/Llama-3.2-1B`
[3]`https://huggingface.co/huggingface-course/bert-finetuned-squad`

## D Experiment Settings

### D.1 Experiment settings for benchmark with all baselines

**Benchmark compression on ResNet models.** We present experiments using ResNet architectures on the CIFAR-10 and CIFAR-100 datasets. When compressing ResNet models, our method requires fine-tuning over the training dataset, completing the compression process within 10 epochs. To achieve high compression rates, we represent each layer's weights with either 4 or 16 components (i.e. $K = 4$ or $K = 16$ for each layer) and apply a sparsity level of 50%. As shown in Table 1, our methods compress the models by factors ranging from $16 \sim 32\times$ while keeping accuracy drops below 1.3%. For example, compressing ResNet-20 by a factor of 16 results in an accuracy drop of only 0.52%. Likewise, compressing ResNet-32 by a factor of $32\times$ yields a minimal accuracy reduction of 1.29%. Additionally, we compress ResNet-56 by a factor of 32, observing an accuracy drop of only 0.84%. Compared to other methods, our approach achieves much higher compression rates with smaller decreases in accuracy.

**Benchmark compression on BERT-base model.** We further investigate our compression method on attention-based models. We apply our compression model on BERT-base (Devlin et al., 2019) model and test it on the SQuAD V1.1 dataset (Rajpurkar et al., 2016). Similarly, we consider the F1 score drop and compression rate as the evaluation metrics. During the compression process, the BERT model is fine-tuned on the training dataset, with the entire procedure completed within 3 epochs.

We compressed the BERT model using $K = 16$ Gaussian components and pruned 75% of its parameters, leading to a $32\times$ compression rate. We employed layer-wise quantization combined with unstructured pruning to attain these results.

**Benchmark compression on Llama and Qwen models.** Due to hardware limitations, we cannot run very large-scale LLMs, which are Llama3.1-8B and Qwen2.5-7B.

### D.2 Experiment settings for ablation studies for SQS method

**Impact of different priors.** For comparison, we consider a zero-mean Gaussian distribution as the prior and replace the delta distribution with a Gaussian distribution in the variational family. That is, any $q'(\theta) \in \mathcal{F}'$ has the form:

$$\tilde{\theta}_i | \gamma_i \sim \gamma_i \sum_{k=1}^{K} \phi_k(\theta_i) \mathcal{N}(\mu_k, \sigma_k^2) + (1 - \gamma_i) \mathcal{N}(0, \sigma_0^2), \qquad \gamma_i \sim \mathtt{Bern}(\tilde{\lambda}_i).$$

Based on this, we can get the modified marginal variational distribution $q'(\tilde{\theta}_i)$ as:

$$q'(\tilde{\theta}_i) = \tilde{\lambda}_i \sum_{k=1}^{K} \phi_k(\theta_i) \mathcal{N}(\mu_k, \sigma_k^2) + (1 - \tilde{\lambda}_i) \mathcal{N}(0, \sigma_0^2). \tag{34}$$

Thus, following the same reasoning and derivation as we get the equation (8), The Gaussian prior can be obtained by:

$$\Omega' = -\mathbb{E}_{q'(\tilde{\theta})} \big[ \log p(D; \tilde{\theta}) \big] + \sum_{i=1}^{T} \mathrm{KL} \big( q'(\tilde{\theta}_i) \| \mathcal{N}(0, \sigma_0^2) \big)$$

$$= -\mathbb{E}_{q'(\tilde{\theta})} \big[ \log p(D; \tilde{\theta}) \big] + \sum_{i=1}^{T} \mathrm{KL} \big( q'(\tilde{\theta}_i) \| (\tilde{\lambda}_i + (1 - \tilde{\lambda}_i)) \mathcal{N}(0, \sigma_0^2) \big)$$

$$\leq -\mathbb{E}_{q'(\tilde{\theta})} \big[ \log p(D; \tilde{\theta}) \big] + \sum_{i} \tilde{\lambda}_i \mathrm{KL} \left( \sum_{k=1}^{K} \phi_k(\theta_i) \mathcal{N}(\mu_k, \sigma_k^2) \Big\| \mathcal{N}(0, \sigma_0^2) \right). \qquad \text{(Gaussian prior)}$$

We compare the impact of the above Gaussian prior with the proposed Spike-and-GMM priors and summarize the result in Table 4.

**Description of Baselines.** For the following lines of baselines, we use the reported results in their papers:

- Optimal Brain Surgeon (OBS) (Hassibi et al., 1993) selects weights for removal from a trained neural network using second-order information.

- BitSplit (Wang et al., 2020a) incrementally constructs quantized values using a squared error metric based on residual errors.

- AdaQuant (Hubara et al., 2021) utilizes STE for direct optimization.

- BRECQ (Li et al., 2021) integrates Fisher information into the optimization process and focuses on the joint optimization of layers within individual residual blocks.

- Exact Optimal Brain Quantization (OBQ) (Frantar & Alistarh, 2022) adapts second-order weight pruning methods to quantization tasks.

- GPTQ (Frantar et al., 2023) employs second-order information for error compensation on calibration sets to speed up generative models.

We adopt their implemented code and use the same setting for training and testing:

- AWQ (Lin et al., 2024) implements activation-aware quantization, selectively bypassing the quantization of key weights[4]. This method is a training-free method, which does not need extra training on the selected dataset.

- DGMS (Dong et al., 2022) is an automated quantization method that utilizes Mixtures of Gaussian to avoid the aforementioned problem[5]. We use their code base and configure it with the same hyperparameters. Their algorithm is trained on the same dataset for fairness of comparison.

**Definition of Evaluation Metrics.** Let $K$ denote the number of shared weight vectors. Then, the metric `Bits` is defined as $\log_2 K$.

## E Long-tail Full-precision Weight Distribution of Llama3.2 and Qwen2.5 Models

We present the visualization of the long-tailed weight distributions for the Llama3.2 model in Figures 6 and 7, and for the Qwen2.5 model in Figures 4 and 5.

---

[4]`https://github.com/mit-han-lab/llm-awq`
[5]`https://github.com/RunpeiDong/DGMS`

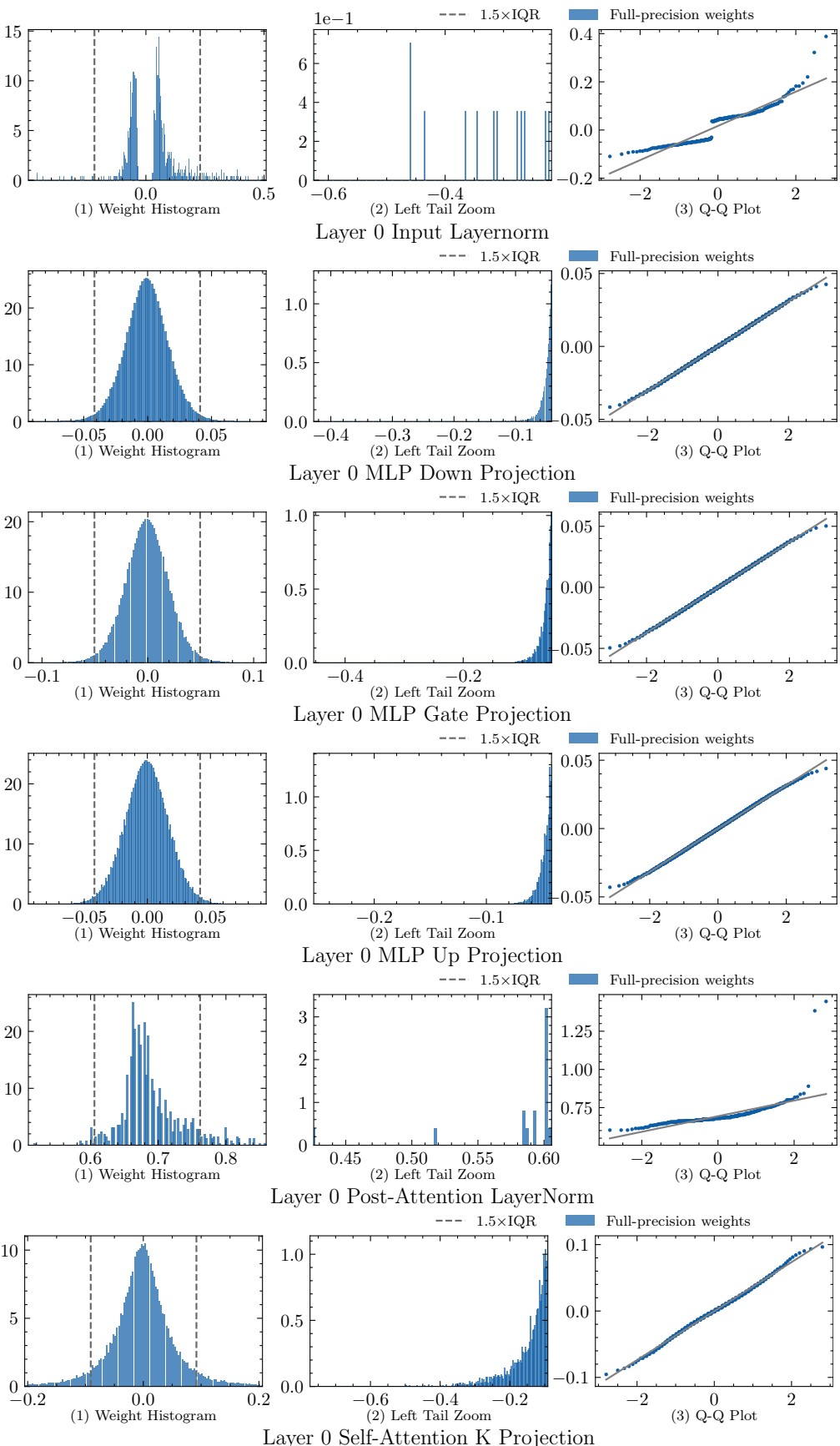

Figure 4: Long-tail weight distribution of different layers in Qwen2.5 model (part 1).

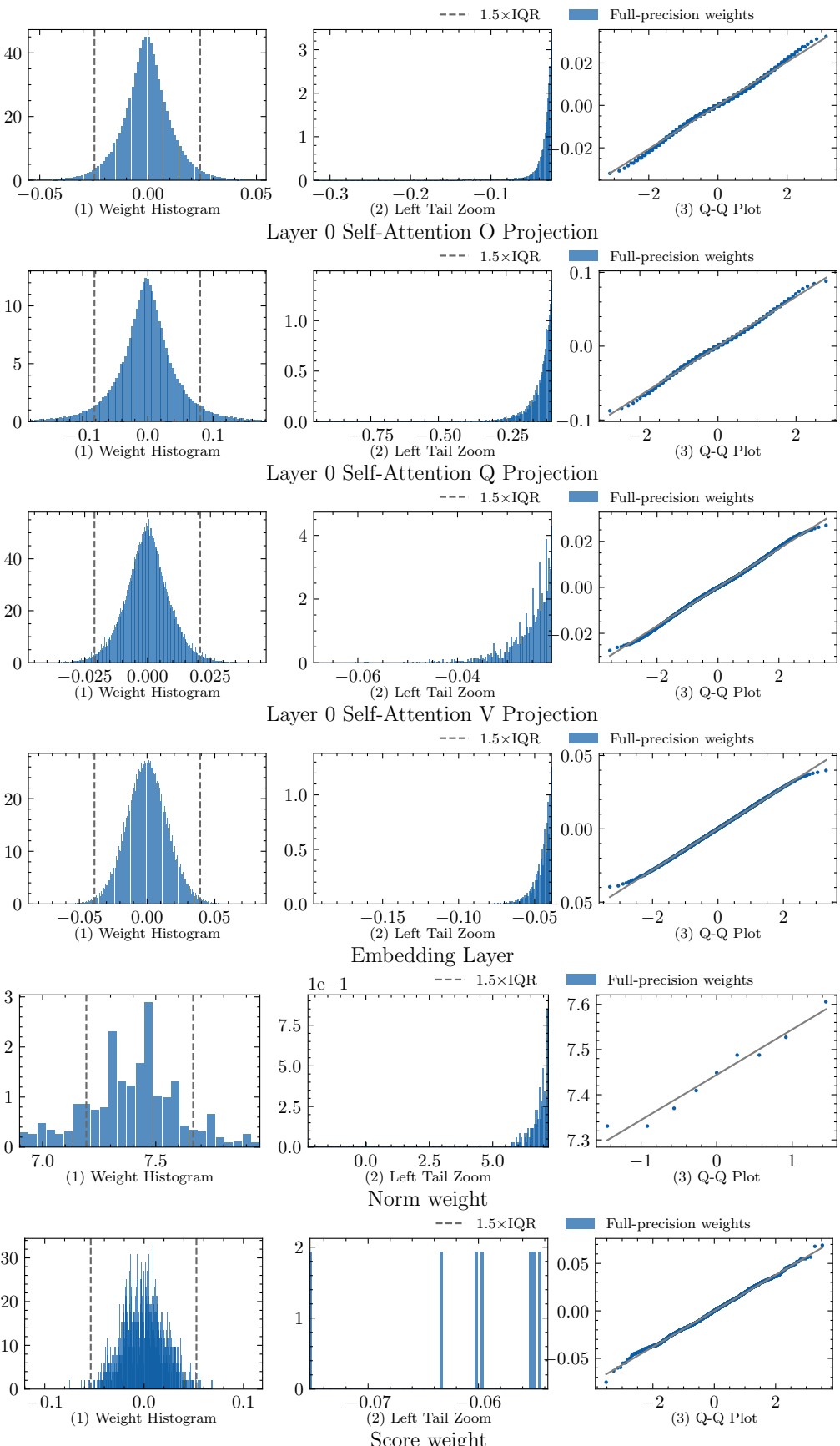

Figure 5: Long-tail weight distribution of different layers in Qwen2.5 model (part 2).

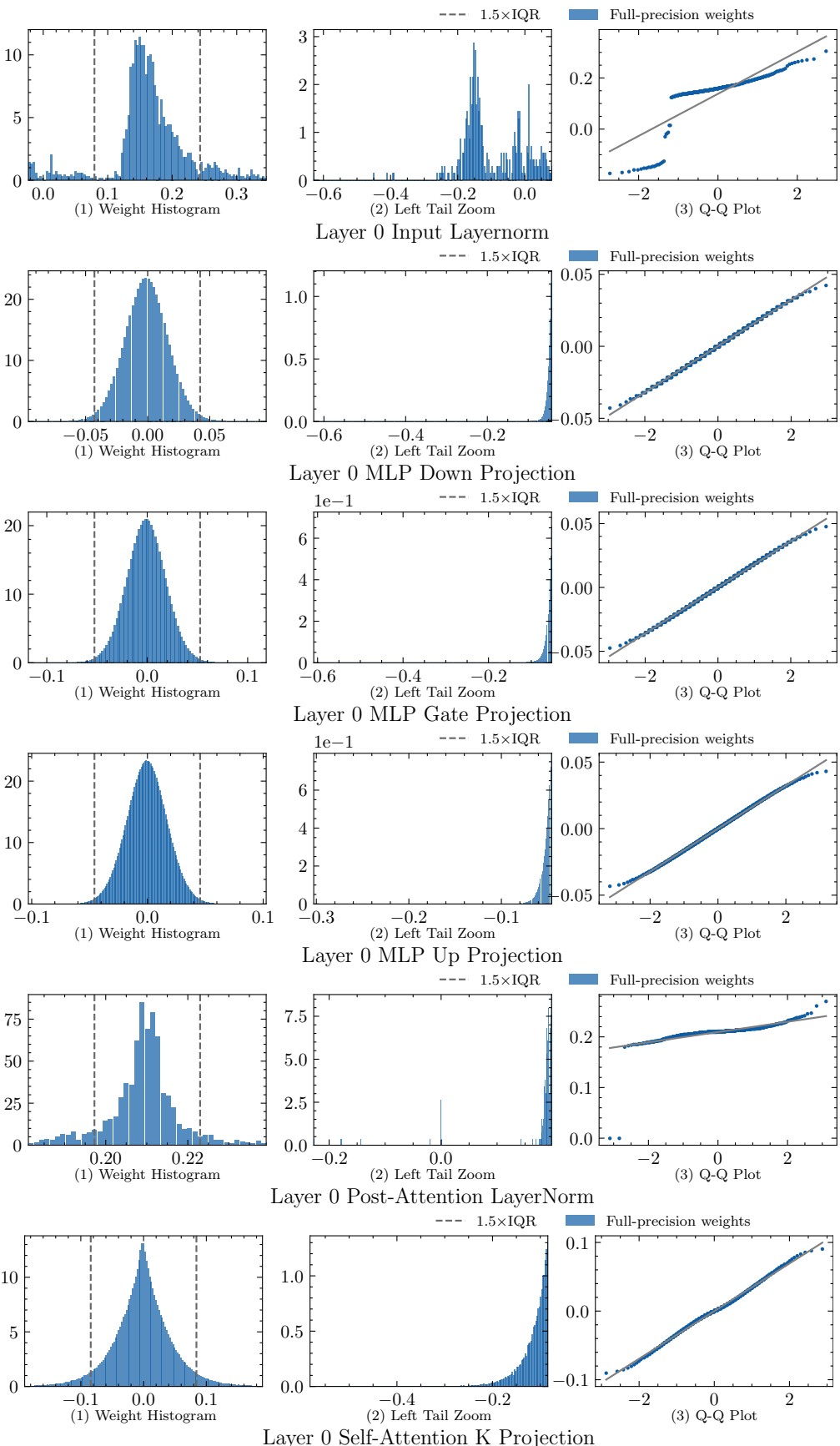

Figure 6: Long-tail weight distribution of different layers in Llama3 model (part 1).

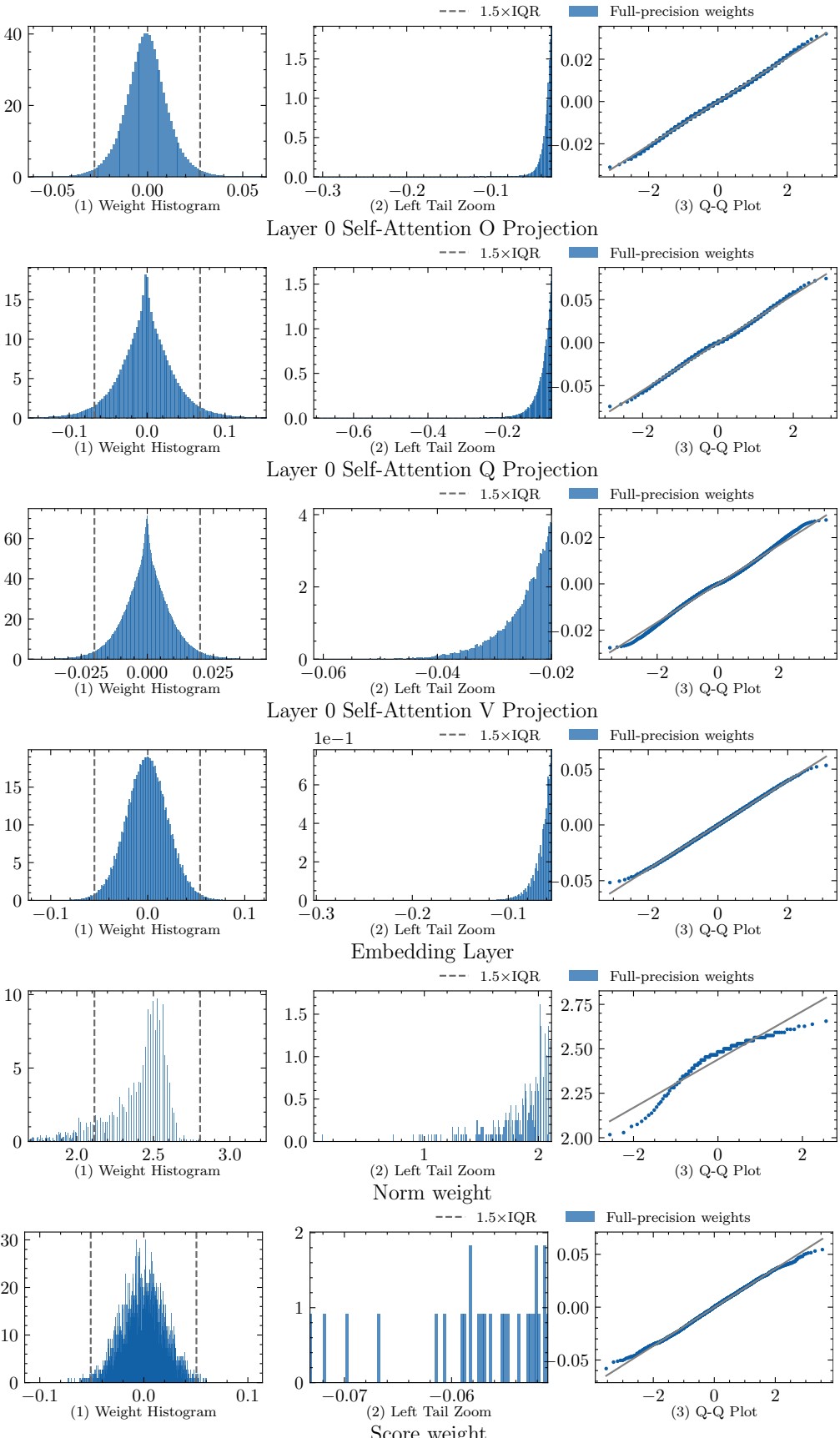

Figure 7: Long-tail weight distribution of different layers in Llama3 model (part 2).

