# OpenReview forum: "SQS: Bayesian DNN Compression through Sparse Quantized Sub-distributions"
_TMLR — Under review for TMLR_

### Review · Reviewer_ZojM · 2026-06-26

**Summary Of Contributions:**

This paper proposes a method for the joint sparsification and quantization of deep neural network models usng approximate variational learning. It extends a previously proposed method for quantizing deep neural networks, differentiable Gaussian mixture weight sharing (DGMS), in two ways. First, it employs a spike-and-slab prior, specifically a spike-and-GMM, in the modeling of the DNN weights to perform joint sparsification and quantization, rather than just quantization as DGMS does. Because computation of the KL divergence between the marginal variational distribution and the prior is intractible, the paper introduces an approximation to the evidence lower bound needed for variational learning. Second, to better handle models like Transformer-based large language models, which have a heavy-tailed weight distribution, the paper proposes a partitioning of weights into four groups using a modified 1.5 x Interquartile Range (IQR) rule. In contrast, the DGMS paper does not consider self-attention models. Third, the paper proposes a Bayesian averaging approach to inference and shows that it outperforms the greedy approach that uses the most likely quantized weights. The paper evaluates the proposed algorithm on various ResNet models, BERT-base, Llama3.2-1B, and Qwen2.5-0.5B.

Key strengths
+ The paper proposes a nice extension to DGMS that allows for joint quantization and sparsification of DNN models. Experimental results show that the method is effective, and ablation experiments illustrate the importance of various design decisions in the proposed algorithm.
+ The paper evaluates the proposed algorithm on both ResNet and self-attention models.
+ The authors are open-sourcing the code associated with the paper, making it easy for others to apply and build on this work.

Key weaknesses
- The discussion of Bayesian averaging is a bit thin and is missing important information.
- The paper preparation could have been a bit more careful. There are a number of typographic errors that should be corrected. They are listed under "Requested Changes".
- The paper buries an important limitation of the LLM quantization experiments in Appendix C, namely the need to specialize the LLM to the task used for quantization: "We then fine-tune the model on the considered SST-2 task before applying our SQS for compression. We find that omitting the fine-tuning step significantly degrades the performance of SQS." This is an important limitation, as it implies that a quantized LLM may perform well only on tasks similar to the dataset used for quantization and sparsification, and it should be explicitly discussed in in a "Limitations" section in the paper.

**Additional Comments:**

None

**Audience:**

Yes

**Audience Explanation:**

As deep learning models are more widely deployed, methods to reduce their storage and inference costs are becoming increasingly relevant. This paper is a solid contribution in this direction.

**Broader Impact Concerns:**

No need for a broader impact statement.

**Claims And Evidence:**

No

**Claims Explanation:**

The bulk of the papers claims are well supported, but not all of them.
- Claim: the proposed algorithm is competitive with existing approaches. The experimental results in Tables 1-3 provide good evidence for this claim, as does the proof of Theorem 1, even if it is limited to fully connected MLPs doing regression.
- Claim: the spike-and-slab prior outperforms a GMM prior. Table 4 makes this case convincingly.
- Claim: the IQR-based partitioning improves the modeling of long-tailed weight distributions. Figures 2 and Figures 4-7 (the latter set in Appendix E) make a good case for this claim. However, it would be more strongly supported if there were an ablation experiment that contrasts the performance of models quantized with and without the partitioning and keeps the total number of Gaussian components constant (meaning, I think that you would need 4K components without the partitioning). It would be good to see the impact on task performance from the partitioning, and not just plots of weight distributions.
- Claim: Bayesian averaging gives better performance than greedy inference using the most likely weights. This claim is partially supported by the results shown in Figure 3; however, _this topic is not explored in sufficient detail_. The number of samples used in Bayesian averaging does not appear to be provided in the paper, and that is important information. It would also be great to see some results showing how performance varies with the number of samples used in the averaging so the reader can understand the tradeoffs between computation and task performance. Also, is it correct to assume that the samples could be precomputed and then used at inference time, so the sampling would be a one-time operation that effectively increases the model size by a factor of M?

**Requested Changes:**

- **Required**: Add at least one LLM quantization and sparsification experiment that compares the performance of models compressed with and without the outlier-aware windowing, but otherwise keeping the degree of compression fixed between the two models, to show that the windowing strategy impacts task performance.
- **Required**: Expand the discussion of Bayesian averaging to provide the number of samples used in the experiments and an explanation of whether or not the weight samples can be precomputed.
- **Optional**: Add one or more experimental results showing how task performance varies with the number of samples used in Bayesian averaging.
- **Required**: Discuss the limitation that LLMs compressed using SQS must be pretrained on the task used for quantization more prominently.
- **Required**: correct the typographic errors listed below.

p. 3, Equation 4: missing $=$ sign

p. 3: "slack-and-slab" => "spike-and-slab"

p. 4: "Please refere to Appendix A." => "Please refer to Appendix A."

p. 5: Missing parentheses around the Dekking (2005) citation due to the use of \cite or \citet instead of \citep.

p. 6: "...regression tasks with fully connected neural networks and shows that the variational posterior of sparse and quantized neural networks, i.e., the optimization of Equation (7)." -- "shows that" what? I think there was an editing mishap on this sentence.

p. 7: "A L-hidden layer fully connected NNs" => "A L-hidden layer fully connected NN"

p. 7: "shows that under $B \geq 2$" -- $B$ is only discussed in the appendix, so this statement is confusing.

p. 19: Missing parentheses around the Hershey & Olsen (2007) citation due to the use of \cite or \citet instead of \citep.

p. 32: "exploit more around the optimal." Do you mean "optimum"?

---

### Review · Reviewer_X1rp · 2026-06-28

**Summary Of Contributions:**

In this work, the authors introduce a framework for simultaneous pruning and quantizing parametric models via Bayesian variational learning. The central idea is to use a spike-and-slab prior to induce sparsity, and the slab part employs a Gaussian Mixture to handle the quantization.
The authors present a series of numerical experiments, mainly for the compression of pretrained vision and language models.
The authors also present a theoretical result regarding the consistency of the method.

**Strengths**
- The proposed method is simple and seems effective from the numerical evidence presented.
- The numerical evidence is, in my opinion, sufficient to justify the claims, despite not covering all interesting aspects of the proposed. method.

**Weaknesses**
- In general, the presentation of the manuscript and the overall writing are not clear. I will clarify all the points below.
- Sometimes the notation is confusing.
- Proof lacks clarity in the chain of arguments. Moreover, the work is missing a clear contribution section and an in-depth discussion about the theoretical results.

**Additional Comments:**

I have an additional question for the authors:
- On page 20, why not use $k^*  =\arg\max_{k} \phi_k(\theta_i)KL(N(\mu_k,\sigma_k^2)|| N(0,\sigma_k^2))$ instead of what you propose ? It seems more natural, but I don't think it is equivalent. I would be grateful if the authors could clarify this point.

**Audience:**

Yes

**Audience Explanation:**

Model compression through sparsity and quantization has had a very important impact on model compression in recent years. Quantization-based methods, in particular, are now ubiquitously used in handling large models in constrained devices. For this reason and for the background of the proposed method, I believe the TMLR community would be interested in this contribution.

**Claims And Evidence:**

Yes

**Claims Explanation:**

Numerical experiments showcase that the proposed framework, despite its simplicity, can match or outperform other compression strategies.
I believe the manuscript proposed by the authors could make a very strong work, conditional on making it more readable

There is, in fact, a lack of organization in the work, which limits the overall clarity of the contribution, both in the main corpus and in the appendix. In particular, for the appendix, for an unfamiliar reader it could be very complicated to follow as the authors provide no proof-line before diving into the technical lemmas needed. Notation is sometimes unclear, e.g., the notation for $q(\tilde \theta_i)$ (which is a distribution in the right-hand side of eq.6, but $\tilde \theta_i$ never appears). This also happens similarly in other points of the paper.
I believe notation and contribution highlighting are the main points strongly compromising clarity right now, together with the totally unguided proof presentation in the appendix.

**Requested Changes:**

- On page 2, the sentence "a general stochastic quantization is a mapping...". I think it would be more formally correct to define $\mathcal Q : \mathbb R \to \mathcal P_K(\mathbb R)$ as a map from the real numbers to the space of probability distributions over $\mathbb R$ with finite support of cardinality $K$. Moreover, I think it is with "with probability $p_{k_i}$ and not $p_{ki}$.
- On page 3, following standard notation, the likelihood is usually denoted with $p(D|\theta)$.
- On page 4, there is a typo in "Please refere to Appendix A. for a detailed derivation of Equation (8)."
-  On page 5, "learning, associated with the optimal parameter estimations $\{\hat{\theta_i}, \hat{\mu_i}, \hat{\sigma^2_i}, \hat{\lambda_i}\}$". Isn't $\theta$ fixed from the beginning as trained parameters? I think it is a bit confusing to include it in the model parameters to fit.
- Page 7, what is the notation $\log^\delta$? It was never introduced in the main text until now, and it's not standard.
- The related works section is strangely positioned. I think it could be beneficial to adapt it and position it after the introduction, so that a reader gets a clear idea of the state of the art before diving into modeling details.
- In Theorem 3 in the appendix, "\varepsilon_n^*" is used and defined for the first time later (on Lemma 5).
- I would suggest the authors include a clear bullet-point paragraph clarifying precisely which are the contributions of the work. Unfortunately, the lack of discussion on the theoretical result and the too dense and not guided proof-line in the appendix, makes it difficult to identify what exactly is novel from the theoretical side.
- Make notation coherent and clear. If an object is a distribution, then on the right-hand side, there must be a distribution. If you want to specify which object's distribution you are talking about, it can be put as a pedix, for example, or any other way that makes it clear).

---

### Review · Reviewer_fjvi · 2026-07-04

**Summary Of Contributions:**

- Instead of applying pruning and quantization sequentially, the authors introduce a unified framework that models network weights using a combined spike-and-slab prior and a Gaussian Mixture Model (GMM) posterior.
- The paper tackles a significant and practical challenge: compressing large-scale DNNs and LLMs for deployment on resource-constrained edge devices.
- The proposed mathematical framework is principled, successfully stringing together variational inference techniques to make the joint optimization of discrete quantization and pruning tractable and scalable.
- The authors provide a rigorous theoretical proof demonstrating that the variational posterior of the sparse and quantized network converges asymptotically to the true underlying target network, adding valuable statistical guarantees to the method.
- The integration of Bayesian Model Averaging (BMA) over the learned sparse quantized sub-distributions is a well-motivated approach to effectively smooth out quantization noise and preserve model accuracy in extreme compression regimes.

**Audience:**

Yes

**Audience Explanation:**

Yes, at least some individuals in TMLR's audience would be highly interested in knowing the findings of this paper.

Given the research shift toward accelerating generative transformers (like LLaMA and Qwen), researcher community studying low-bit quantization sensitivity would be very interested in the paper's findings.

Another interesting fact is the strong theoretical proofs with empirical evidence makes this paper very interesting for research folks interested in fundamental understanding of quantization and pruning.

**Broader Impact Concerns:**

The authors must expand their concluding remarks or add a dedicated Broader Impact Statement that acknowledges training-to-deployment trade-offs, including inference latency, and safety issues (if any).

**Claims And Evidence:**

Yes

**Claims Explanation:**

Although paper provides the evidence, the evidence is not yet clear or convincing enough to validate the paper's broad claims.

- The authors claim that SQS achieves significantly higher compression rates than AWQ while maintaining comparable performance drops on LLMs. However, the evidence supporting this claim is not convincing to me. The experimental setup described in Appendix C shows that SQS utilizes extensive task-specific supervised fine-tuning on the SST-2 dataset, whereas the AWQ baseline is training-free, zero-shot PTQ approach. It is not clear whether the superior performance of SQS is due to the effectiveness of the compression algorithm itself or simply the result of target task fine-tuning.
- A core claim of the paper is that SQS is suitable for deploying models on resource-constrained devices. Yet, the primary inference mechanism proposed to maintain accuracy is BMA. Eq. 10 shows that the final prediction requires averaging over $M$ separate sampled weight configurations. This means running $M$ full forward passes per single prediction, which linearly multiplies inference FLOPs, latency, and operational memory overhead. Please consider discussing how this linear increase in computational latency and memory overhead impacts the deployment on resource-constrained devices, and clarify whether this multi-pass approach was used in the benchmark tables, which could affect compute parity against single-pass baselines.
- The evaluation primarily benchmarks SQS against methods that perform pruning and quantization sequentially or separately. To better demonstrate performance, the authors might consider comparing against established joint optimization frameworks. (e.g., https://arxiv.org/abs/2005.07093)
- Eq. 16 calculates the compression rate based on the codebook and the quantized indices of non-zero weights but appears to omit the structural memory overhead required to store the sparsity mask. Accounting for this structural overhead is critical for a fair comparison against dense quantization baselines.

**Requested Changes:**

Please see the empirical evidence part above and fix all the suggestions provided there.

---

### Comment · Reviewer_ZojM · 2026-07-20
**No revision or response from the authors?**

I was surprised to get the request for a final recommendation on this paper given that I have not seen any response from the authors to the reviews.

Dear authors, are you planning to reply?

Thanks,

Reviewer